# Constraining microphysics assumptions on the modeling of Atmospheric Rivers using GNSS Polarimetric Radio Occultations

Antía Paz[1,2,3], Ramon Padullés[1,2], and Estel Cardellach[1,2]

[1]Institut de Ciències de l'Espai, Consejo Superior de Investigaciones Científicas (ICE-CSIC), Barcelona, Spain
[2]Institut d'Estudis Espacials de Catalunya (IEEC), Barcelona, Spain
[3]Facultat de Física, Universitat de Barcelona (UB), Carrer de Martí i Franquès, 1-11, 08028 Barcelona, Spain

**Correspondence:** Antía Paz (a.paz@csic.es)

**Abstract.** Improving the representation of precipitation in weather and climate models remains a major challenge, particularly due to uncertainties in microphysical parameterizations that govern hydrometeor characteristics and behavior. The Polarimetric Radio Occultation (PRO) technique enhances the standard Radio Occultation (RO) method by offering vertical profiles of both precipitation structure and thermodynamic atmospheric variables. PRO achieves this by utilizing two orthogonal polarizations—horizontal (H) and vertical (V)—to measure the differential phase shift ($\Delta\Phi$), which represents the difference in phase delay between the two of them. This study focuses on assessing the sensitivity of the PRO technique to the vertical structure of hydrometeors under different microphysical assumptions. To explore this sensitivity, simulations are conducted using the Weather Research and Forecasting (WRF) model, with particular attention to the effects of different microphysics schemes on the simulated $\Delta\Phi$. The study also incorporates the Atmospheric Radiative Transfer Simulator (ARTS) particle database to characterize hydrometeors based on their scattering properties. Atmospheric Rivers (ARs) are used as a case study. The $\Delta\Phi$ values simulated under different microphysics schemes are compared to GNSS-PRO observational data from PAZ and Spire satellites, providing a means to evaluate the performance of the WRF microphysics parameterizations. A simplified forward operator is built in terms of water content (WC) and x-parameter (factor relating WC and specific differential phase shift, $K_{dp}$) , and the best value for $x$ is obtained through optimization techniques after comparing with actual $\Delta\Phi$ observations. The lowest cost function values systematically correspond to Goddard and WSM6 microphysics (75% of the cases). Similarly, the best choice for the x-parameter values approach $\sim 0.1 mm/kg \cdot m^{-2}$. Once compared to the results from the ARTS particle database, the particle habits that best represent such relationship are bullet rosettes and snow aggregates.

## 1 Introduction

The Global Navigation Satellite System (GNSS) Radio Occultation (RO) technique is a remote sensing method that tracks signals emitted by navigation satellites (e.g. GPS, Galileo, etc.) from a Low Earth Orbit (LEO) satellite as it rises or sets behind the Earth's limb. This method enables the retrieval of atmospheric refractivity by analyzing the delay and bending of radio signals as they propagate through the atmosphere. As these signals traverse increasingly dense atmospheric layers, they experience bending caused by vertical gradients of the refractive index. This delay can be used to derive profiles of radio refractivity and ionospheric total electron content. From these refractivity profiles, key atmospheric variables such as

pressure, temperature, and water vapor pressure can be retrieved, spanning from the stratosphere to the surface with a vertical resolution of 100–300 meters (e.g., Kursinski et al., 1997). These atmospheric products are operationally assimilated into global Numerical Weather Prediction (NWP) models (e.g., Healy et al., 2005).

GNSS systems, such as GPS, transmit signals in a Right Hand Circularly Polarized (RHCP) state. While standard Radio Occultation (RO) receivers are designed to capture this polarization, the Polarimetric Radio Occultation (PRO) technique extends this capability by receiving at dual orthogonal polarizations, horizontal (H) and vertical (V). This setup enables the measurement of the differential phase delay ($\Delta\Phi$, defined in Section 2.1 below), which corresponds to the phase difference between the horizontally and vertically polarized components. This differential delay is induced when the GNSS signals propagate through nonspherical hydrometeors, raindrops or ice particles, which alter the signal path due to their shape and orientation (Cardellach et al., 2015). Through this mechanism, PRO provides additional sensitivity to the vertical structure of precipitation, offering a valuable enhancement to traditional RO observations, becoming the first technique to provide both types of observations simultaneously from the same instrument.

Operational testing of the PRO technique began in 2018 with the Spanish LEO satellite PAZ as part of the Radio Occultation and Heavy Precipitation (ROHP) experiment, led by the Institut de Ciéncies de l'Espai-Consejo Superior de Investigaciones Científicas/Institut d'Estudis Espacials de Catalunya (ICE-CSIC/IEEC) in collaboration with the NOAA, UCAR, and NASA/JPL. Since 2023, the technique has also been implemented on several Spire Global CubeSats (Talpe et al., 2025), two PlanetiQ satellites, and one Yunyao Aerospace satellite (Kan et al., 2025). PRO's primary objective is the detection of intense precipitation, a capability that has been successfully demonstrated (Cardellach et al., 2019). Furthermore, it has been shown that PRO is sensitive to both heavy rainfall and horizontally oriented frozen hydrometeors (Padullés et al., 2021, 2023). Validation of the $\Delta\Phi$ observable has been performed using two-dimensional datasets such as the Integrated Multi-satellite Retrievals from GPM (IMERG) (Cardellach et al., 2019; Padullés et al., 2020). Passive microwave radiometers have also been utilized to interpret precipitation vertical structures (Turk et al., 2021). More recently, three-dimensional data from the Next Generation Weather Radars (NEXRAD) have been used to assess the vertical structure of the measured $\Delta\Phi$ (Paz et al., 2024).

Accurately representing the vertical structure and intensity of precipitation in models remains a significant challenge due to uncertainties in microphysical assumptions. These uncertainties strongly affect the simulation of hydrometeor populations and their associated polarimetric signatures, as demonstrated by recent studies (e.g. David et al. (2025)). One of the key limitations lies in the treatment of microphysical processes, which are often represented through parameterizations that vary significantly across models and resolutions. In current cloud-resolving simulations, the introduction of hydrometeors and the associated latent heat release are handled using convective and/or microphysical parameterizations, both of which carry large uncertainties, (e.g. Hristova-Veleva et al. (2021)). PRO offers a unique opportunity to constrain these assumptions by providing vertically resolved information on hydrometeor and thermodynamic characteristics, complementing existing observational methods. This study aims to assess the use of PRO in constraining microphysical assumptions within models, exploiting the sensitivity of the technique to the amount of water content and to particle shapes. These changes depend on the assumed microphysics, and may lead to differences in the forecast of relevant magnitudes such as the amount of rainfall. To achieve the objective, simulations are conducted using the Weather Research and Forecasting (WRF) model with different microphysical

parameterizations. The WRF model provides a robust framework for simulating atmospheric phenomena with the flexibility to adjust parameterizations representing key processes within these events (Skamarock et al., 2019). This study focuses on evaluating how different microphysical parameterizations affect cloud microphysics and, in turn, the simulated $\Delta\Phi$. Because these small-scale processes are not explicitly resolved in numerical models, their representation relies on assumptions that can vary significantly across schemes. By analyzing how these assumptions impact the simulated differential phase shift, we aim to assess whether PRO observations can distinguish between different microphysical treatments, an idea previously suggested by Murphy et al. (2019) based on idealized simulations.

To simulate the $\Delta\Phi$ observable from PRO, it is necessary to quantify the electromagnetic scattering properties of hydrometeors. However, the microphysical schemes used in mesoscale models like WRF do not explicitly provide the scattering properties required to compute the specific differential phase shift ($K_{dp}$) at GNSS frequencies (around 1.4 GHz) and for limb-viewing geometries. To address this limitation, we first simulate $\Delta\Phi$ using a simplified operator that uses water content (WC) to derive $K_{dp}$ through a linear relationship, through a factor that we define as x-parameter. An optimization process is run on the x-parameter so that the one that minimizes a cost function is chosen as the proportionality factor. Then, following Padullés et al. (2025), the Atmospheric Radiative Transfer Simulator (ARTS) habit database is incorporated so that the different habits are checked against the best x-parameter. The use of the ARTS database also ensures consistency with other scientific communities, such as the microwave (MW) remote sensing community, which employ these parameters for assimilation into NWP models, (Geer et al., 2021). Also, the coupling of WRF and ARTS has already been used in other studies demonstrating its validity to represent satellite observations (e.g. Wang et al. (2016)).

Our study will focus on Atmospheric Rivers (AR). These events are narrow, elongated corridors of concentrated atmospheric moisture transport, often originating in the tropics and extending towards mid-latitudes. They play a critical role in the global water cycle, delivering substantial precipitation when the moisture interacts with topographic features or frontal systems. While ARs contribute to beneficial water supplies, they are also associated with extreme rainfall and flooding, being also responsible of important damage and loss. Their spatial extent enables a large amount of coincidences with PRO observations and allows a systematic study that comprises around 40 cases split in two different regions: the north-east Pacific and the north Atlantic. ARs have been widely studied using standard RO (e.g. Murphy and Haase, 2022) and airborne RO (e.g. Haase et al., 2021), with focus on assessing the assimilation of the standard products in their forecast. The study Hotta et al. (2023) first simulated PRO data in several AR cases, demonstrating that these events are good choices of weather events to examine. Our efforts in this analysis extend and build on an earlier effort by Chen et al. (2025), which explores a similar study taking Tropical Cyclones (TC) as case studies.

The primary objectives of the study are then: (1) to study the contribution of hydrometeors to the WC and their influence on $\Delta\Phi$; (2) to evaluate the performance of different microphysics schemes based on the GNSS-PRO observational data; (3) to characterize and identify which general classes of particle habits are consistent with the observations; (4) to investigate potential differences between ARs in the North Atlantic and North-East Pacific regions.

This paper is organized as follows: Section 2 describes the data and methodology used in the analysis; Section 3 presents the results of comparisons with PRO observations and the consequent discussion; and Section 4 presents the conclusions.

## 2 Data and methodology

This section outlines the data sources used for the analysis, including PRO data, WRF outputs, and the ARTS dataset. Additionally, the methodology for defining the forward operator used to calculate the simulated differential phase shift, $\Delta\Phi_{sim}$, will be explained.

### 2.1 Polarimetric Radio Occultation data

In this analysis, we use PRO data obtained from the PAZ satellite and from the Spire nanosatellites.

As noted previously, the PRO technique enables us to compare the phase delay ($\Phi$) corresponding to two distinct measured polarizations. During heavy precipitation events, large, horizontally oriented hydrometeors result in a positive accumulated differential phase shift, $\Delta\Phi = \Phi_H - \Phi_V$, due to the depolarization effect (Padullés et al., 2021).

The differential phase contribution along the propagation path is quantified by the specific differential phase. Since the GNSS

community generally uses length units (mm-shift/km-rain) rather than radians, $K_{dp}$ is expressed in these units and multiplied by $\lambda/2\pi$. The equation for $K_{dp}$ is:

$$K_{dp} = \frac{\lambda^2}{2\pi} \int \Re\{f_H(D) - f_V(D)\}N(D)dD \tag{1}$$

here, $\lambda$ is the wavelength associated with GNSS; $\Re$ represents the real component; $f_H(D)$ and $f_V(D)$ are the forward scattering amplitudes, characterizing how GNSS waves scatter on hydrometeors for horizontal and vertical components, respectively; $D$

is the drop's equivalent diameter; and $N(D)$ refers to the particle size distribution (PSD). The scattering amplitudes are specific for each type and shape of the particles.

The cumulative effect of hydrometeors along the ray path is then given by:

$$\Delta\Phi = \int_L K_{dp}(l)dl \tag{2}$$

In this expression, $\Delta\Phi$ is in units of mm, $K_{dp}$ is in mm/unit-length and $L$ represent the ray path length.

Calibrated PRO profiles from PAZ are available from May 2018 onward (Padullés et al., 2024). Each file corresponds to a PAZ observation and contains the vertical profile of the observable differential phase shift, $\Delta\Phi$, in units of length (mm) and relative to the tangential height of each PRO ray. Similarly, PRO data from Spire is available between May and October 2025, and has undergone similar calibration as the PAZ data (Padullés et al., 2025a). Refractivity gradients cause bending, resulting in rays becoming tangent to the surface at their lowest point, termed the tangent point, $h_t$. The coordinates (latitude and

longitude) for each PRO observation are based on this tangent point with $h_t = 4$ km. Though each ray relates to its tangential height, hydrometeors along its path contribute to the $\Delta\Phi$ value, regardless of altitude, since it is an integrated magnitude as we saw on Equation 2.

Additionally, the PRO files include the geographic positions (latitude, longitude, and altitude) of each ray path between GPS and PAZ, determined through ray-tracing. These locations are calculated and re-gridded so that only rays with a tangent height

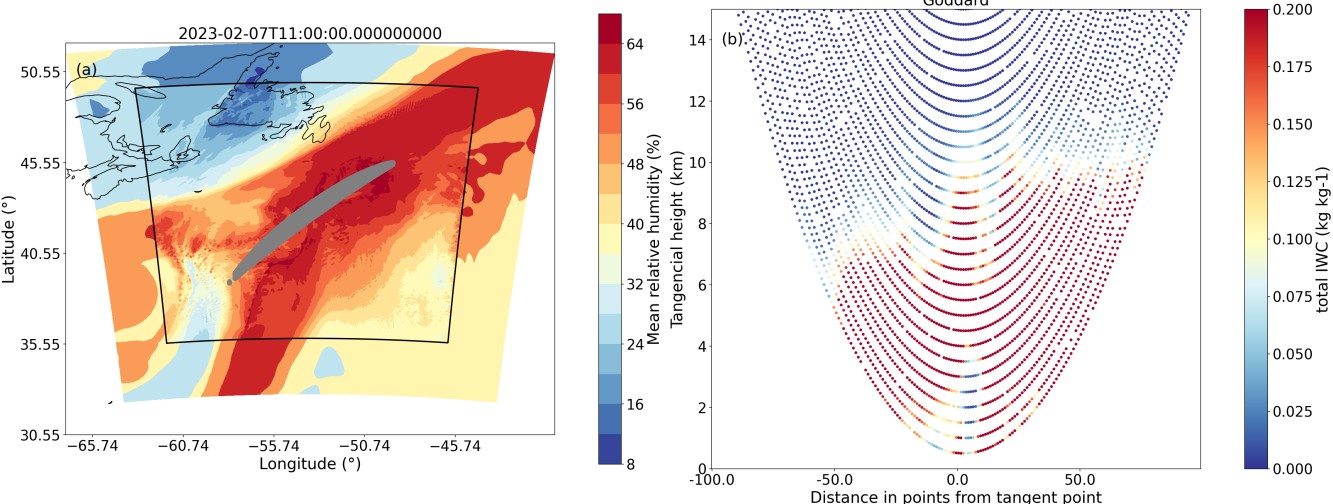

**Figure 1.** Panel (a) shows the mean relative humidity of an AR studied in this analysis, showing the two domains used in WRF. The grey area represents the 2D projection of the PRO rays below 20km. Panel (b) displays some of the occultation rays associated with this observation, showing the interpolated field of snow water content. The roid associated with the case is PAZ1.2023.038.10.59.G19 , corresponding with the observation in Figure 8 (b).

matching a regular grid between 0 and 20 km, with a vertical resolution of 0.1 km, are included. This approach ensures that ray trajectories align with the vertical resolution of $\Delta\Phi$ and represent the paths contributing to each $\Delta\Phi$ measurement. Each observation profile, whether from PAZ or Spire, has an identifier that we will refer to as "roid" from this point forward.

## 2.2   WRF simulations

The Weather Research and Forecasting-Advanced Research Weather (WRF-ARW) model is employed to simulate the ARs
selected for the study. We use ERA5 reanalysis data for both initial and boundary conditions (Hersbach et al., 2020). To ensure the development of realistic atmospheric structures, the simulation is initialized 12 hours before the PRO observation time, following best practices for model spin-up (Skamarock et al., 2019). The simulations are performed with two nested computational domains: a parent domain with a horizontal resolution of 15 km and a nested domain at 3 km resolution. A cumulus parameterization scheme is applied exclusively in the parent domain to account for unresolved convective processes
(Skamarock et al., 2019). The model configuration featured 52 vertical layers, allowing for high-resolution vertical profiling, and a time step of 90 seconds for the integration.

An example of how the domains for the different ARs considered are defined can be seen in Figure 1 (a). The simulations are structured such that the PRO observation point serves as the center of both domains, ensuring sufficient space to include all ray points below 20 km. Figure 1 (b) also shows the geometry of the occultation rays for that specific observation, displaying
the total water content (rain + snow + ice + graupel), but serves as a reference of the typical geometry.

**Table 1.** Physics schemes used in both domains for all the simulations performed with WRF.

| | Domain 1 (15km) | Domain 2 (3km) |
|---|---|---|
| Cummulus | Kain-Fritsch | — |
| Shortwave and Longwave radiation | RRTMG | RRTMG |
| PBL | Yonsei University Scheme (YSU) | Yonsei University Scheme (YSU) |
| Land Surface | Unified Noah Land Surface Model | Unified Noah Land Surface Model |
| Surface Layer | MM5 Similarity Scheme | MM5 Similarity Scheme |

Four simulations were conducted for each AR, differing only in the microphysics schemes used: the Goddard scheme (Lin et al., 1983; Rutledge and Hobbs, 1984), the Thompson scheme (Thompson et al., 2008), the WSM6 scheme (Chen and Sun, 2002; Hong et al., 2004), and the Morrison two-moment scheme (Morrison and Pinto, 2005). This diverse selection of microphysics parameterizations enables the evaluation of the sensitivity of AR simulations to different treatments of cloud microphysical processes. The rest of the schemes used are shown in Table 1.

In a limited number of cases, the New Goddard Shortwave and Longwave radiation schemes are used in place of RRTMG due to technical issues. Both schemes are widely tested in WRF and provide consistent radiative forcing. For each observation, the four simulations done with different microphysics are generated with the same radiation scheme. Taking into account that the amount of observations with this mis-match is low compared to the total amount of cases, we do not consider that this is a significant aspect regarding the evaluation of the final results.

The four different microphysics schemes employed in this study are Bulk Microphysics Schemes (BMS), which parameterize precipitation particles as "bulk" quantities categorized into distinct hydrometeor classes, such as cloud water, rain, snow, ice, and others. In contrast, an alternative approach, known as Bin Microphysics Schemes, explicitly resolves the size distribution of cloud and precipitation particles by discretizing them into predefined "bins." Due to their computational efficiency, bulk schemes are widely utilized in operational weather forecasting and climate modeling.

The four schemes employed in the study predict five hydrometeor types: cloud water, rain, snow, ice, and graupel. The primary distinctions between these schemes lie in the specific microphysical processes and interactions governing the evolution of these hydrometeors, and also the different assumptions made around the PSD and density. Below, we present a brief description of the schemes, highlighting some characteristics in comparison to other schemes or previous versions of the same.

– **Goddard**: This microphysics scheme is mainly based on Lin et al. (1983) with additional processes from Rutledge and Hobbs (1984). It has an option to choose whether hail or graupel is the third class of ice, however, in this analysis the Goddard 3ICE version with graupel is used. A saturation technique (Tao et al., 1989) is introduced to prevent supersatu-

ration or subsaturation at grid points that are clear or cloudy, respectively. Additionally, microphysical processes related to melting, evaporation, or sublimation are calculated based on the thermodynamic state, ensuring uniform treatment. Lastly, the total mass loss of any species will not exceed its available mass, ensuring a balanced water budget, (Tao et al., 2009; Skamarock et al., 2008). The particle size distributions (PSDs) assumed are modified gamma distributions with shape parameter $\mu = 0$ (i.e., exponential distributions), as in Eq. 3.

$$N(D) = N_0 D^\mu e^{-\lambda D} \tag{3}$$

where $N(D)$ represents the number concentration, $D$ is the diameter, $N_0$, $\lambda$ and $\mu$ are the intercept, slope and shape parameter.

- **WSM6**: This scheme is described in Hong and Lim (2006) and accounts for the same predicted variables as in WSM5 but includes graupel and its associated processes described in Chen and Sun (2002). The ice-related microphysics are the ones proposed by Hong et al. (2004). From the WSM scheme series, the WSM6 is the most suitable for cloud-resolving grids (Skamarock et al., 2008). The hydrometeors follow PSDs represented as exponential distributions.

- **Morrison 2-moment**: The main difference between this scheme and the others is the two-moment part. One-moment schemes only predict the mixing ratios of the hydrometeor species; while two-moment schemes also predict the number concentration of each hydrometeor. However, cloud water is not included in the two-moment part. This scheme is based on Morrison et al. (2005) and Morrison and Pinto (2006). It has also an option for choosing either graupel or hail, but in this analysis we have used the graupel option. The two-moment part allows for a more robust treatment of the size distribution enabling a better calculation of the microphysical process rates and evolution (Skamarock et al., 2008). The PSDs of the hydrometeors are represented as modified gamma distributions with $\mu = 0$ (exponential distributions).

- **Thompson**: Details of this scheme are in Thompson et al. (2008). Unlike other bulk schemes, in this approach, snow size distribution depends both on ice water content and temperature, and is represented using the PSD described at Field et al. (2005) rather than a simple exponential distribution. Also, snow is considered non-spherical and its density varies inversely with diameter. This scheme employs the two-moment configuration for rain and ice, while for the other hydrometeors it is one-moment. For hydrometeors other than snow, the PSD is assumed to be a modified gamma distribution with $\mu = 0$.

For doing the different simulations we have selected different AR that are coincident in space and time with PAZ and Spire observations. In total we have 32 co-locations between ARs and PAZ and 5 between ARs and Spire. In Figure 2 we present the location of the coincidences between ARs and PRO data from PAZ and Spire.

Likewise, the different ARs are classified according to their location. Of the 37 case studies we have, 20 correspond to north Atlantic ARs, and the other 17 to north-east Pacific ARs.

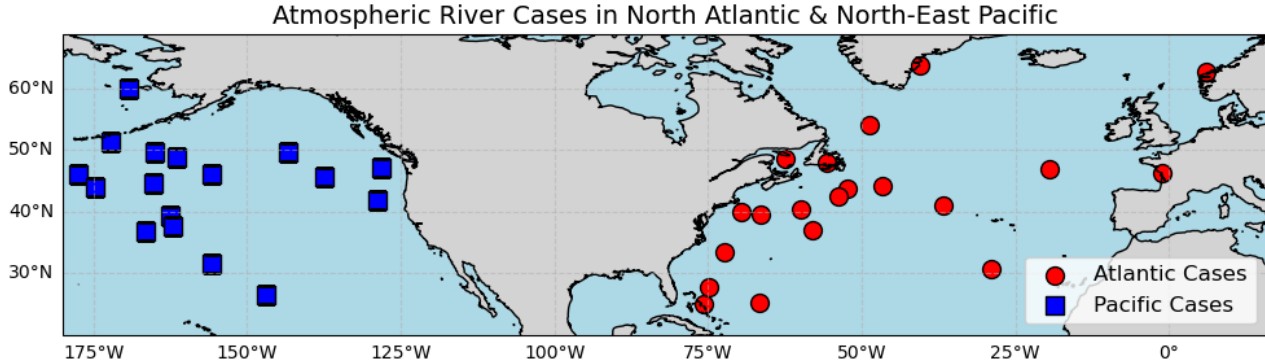

**Figure 2.** Coincidences between observations from PAZ and Spire with ARs. Each point represents the location of the PRO observation coincident with an AR.

### 2.3 Evaluation of Atmospheric River Simulations

The AR cases selected for this study are obtained from the database Guan and Waliser (2024), where we have chosen different
phenomena coincident with PRO observations.

Before analyzing the impact of different schemes and particle habits on the observable $\Delta\Phi$ we perform a brief evaluation of the performance of the WRF simulations in reproducing the overall structure of ARs. This ensures that the simulations provide a realistic meteorological context before assessing microphysical sensitivities. For doing so, we present figures showing the Integrated Vapor Transport (IVT) and the acummulated total precipitation for each of the microphysics schemes, see Figure
3. In addition, the IVT from ERA5 reanalysis is shown and also the infrared temperature image on Figure 4, to provide an observational reference of the large-scale cloud structure of the AR.

The comparison between Figure 3 and Figure 4 indicates that all four microphysics schemes successfully capture the large-scale structure and intensity of the event, including the position, orientation of the moisture plume, and the magnitude of the IVT. This provides confidence in the realism of the simulated atmospheric features. In addition, the accumulated precipitation
plots reinforce this finding, as they exhibit similar spatial patterns and plume alignment across all schemes. However, some differences can be observed in both the total amounts and the spatial distribution of precipitation. For instance, the Goddard and WSM6 schemes tend to produce more localized precipitation maxima, whereas Thompson and Morrison yield a broader, more distributed accumulation. Also, as it will be discussed in the Results section, differences between schemes become more pronounced when analyzing the vertical distribution and water content of the different hydrometeors (rain, snow, ice, and
graupel), having a more direct impact on the simulated $\Delta\Phi$.

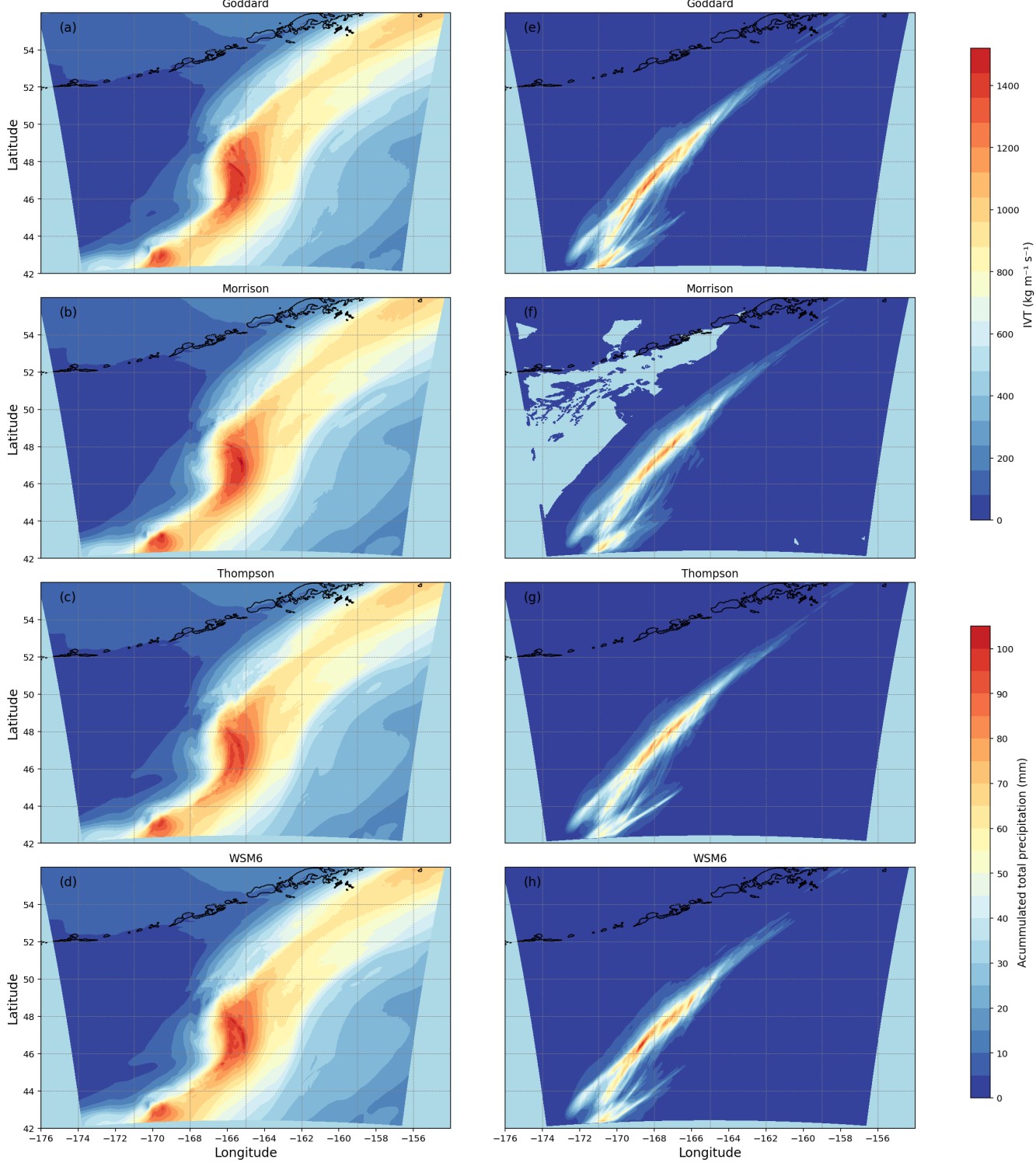

**Figure 3.** Panels (a), (b), (c) and (d) show the IVT from WRF for an specific AR depending on the microphysics scheme employed. Panels (e), (f), (g) and (h) show the accumulated total precipitation for the same schemes. The associated roid with this case is PAZ1.2018.239.03.26.G08.

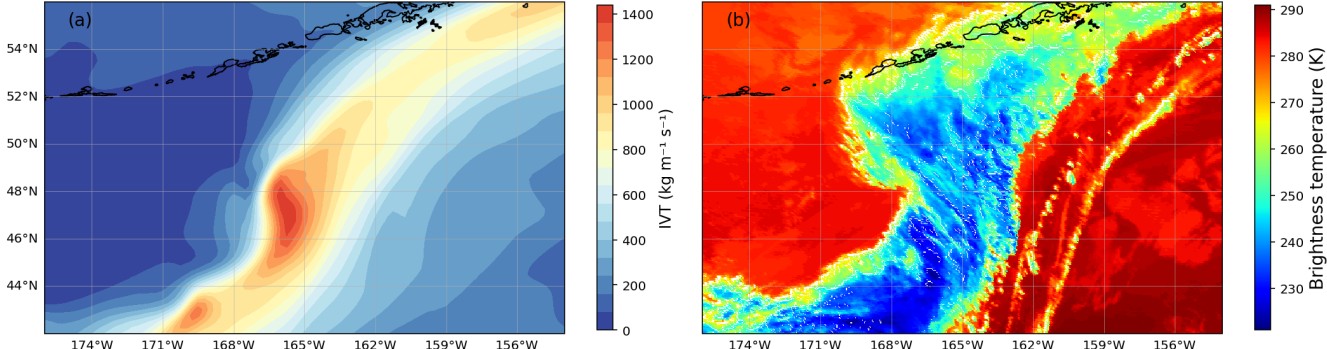

**Figure 4.** Panel (a) shows the IVT from ERA5 reanalysis and panel (b) the associated longwave IR brightness temperature image from geostationary satellite data for the event associated with the observation PAZ1.2018.239.03.26.G08.

## 2.4 $\Delta\Phi$ operator

To compare the vertical profiles of $\Delta\Phi$ observed by PAZ and Spire with those simulated using the WRF model, we develop an observation operator. This operator computes $K_{dp}$ using the WC derived directly from WRF simulations and follows the approach described in Padullés et al. (2025), where a linear relationship between $K_{dp}$ and WC is demonstrated, with the proportionality factor (slope) defined as the x-parameter, $K_{dp} = x \cdot WC$. This relationship is independent of any specific assumptions regarding the particle size distribution or particle habit and forms the basis of the minimization process used later to retrieve the optimal x-parameters.

Then, we defined our observation operator as:

$$K_{dp} = K_{dp,rain} + x_{snow} \cdot WC_{snow} + x_{ice} \cdot WC_{ice} + x_{graupel} \cdot WC_{graupel} \tag{4}$$

The simulated differential phase shift $\Delta\Phi_{sim}$ is then computed as:

$$\Delta\Phi_{sim} = \int K_{dp}dl = \int K_{dp,rain}dl + x_{snow} \cdot \int WC_{snow}dl + x_{ice} \cdot \int WC_{ice}dl + x_{graupel} \cdot \int WC_{graupel}dl \tag{5}$$

In this formulation, each hydrometeor individually contributes to the total $K_{dp}$. For all hydrometeors except rain, the contribution is computed as the product of the x-parameter and the WRF-derived WC. The water content is calculated based on the mixing ratios from WRF simulations using the following expression:

$$WC = q \cdot \frac{p}{R_{dry} \cdot T} \tag{6}$$

where $q$ represents the mixing ratio, $p$ is the pressure, $R_{dry} = 287.1 \times 10^{-3}$ kg m$^{-3}$K$^{-1}$ is the specific gas constant for dry air, and $T$ is the temperature. The contribution of rain ($K_{dp,rain}$) in Equation 4 is treated separately, as the ARTS database models rain as liquid spheres. It is computed using the following empirical relationship:

$$K_{dp,rain} = A \cdot WC_{rain}^B, \quad A = 0.13, \quad B = 1.314 \tag{7}$$

This formulation highlights the dependence of $K_{dp,rain}$ on the rain water content, characterized by specific coefficients $A$ and $B$, (Bringi and Chandrasekar, 2001).

Thus, the computed $K_{dp}$ reflects not only the influence of the microphysical parameterizations used in WRF but also the scattering properties assumed for the hydrometeors.

$$\Delta\Phi_{sim} = \Delta\Phi_{rain} + \sum_{sp}(x_{sp} \ \mathrm{iWC}_{sp}) \tag{8}$$

where $sp$ is the specie for the frozen hydrometeors (i.e. snow, ice, and graupel), and iWC is the integrated WC along the PRO ray paths corresponding to each specie.

Equation 8 allows for a detailed assessment of each hydrometeor's contribution (rain, snow, ice, and graupel) to the differential phase shift ($\Delta\Phi$). Each contribution depends on the WC derived from WRF simulations and the corresponding x-parameter, except for rain, which depends solely on WRF.

The methodology involves generating simulated $\Delta\Phi$ profiles for all the water content fields obtained from WRF employing a range of different values for the x-parameters. In other words, the x-parameters that multiply the different water contents are treated as unknowns that then are resolved by solving an optimization problem, for each microphysics scheme. Specifically, for each scheme, an inverse least-squares problem is solved to determine the x-parameters that, in combination with the water content from WRF, minimize the discrepancy between the observed $\Delta\Phi_{obs}$ and the simulated $\Delta\Phi_{sim}$. The cost function

associated with this optimization is defined as:

$$J(x) = (y - H(x))^T R^{-1}(y - H(x)) \tag{9}$$

where $y$ denotes the observation, $H(x)$ is the observation operator (Eq. 8), and $R$ is the covariance matrices of the observation. The final cost function, $J$, obtained for each observation is then divided by the number of points of the specific observation. Here, we do not employ an initial estimate for solving the problem; instead, we perform a scan of the entire reasonable space

of values from the ARTS particles.

To constrain the problem, specific bounds were set for each hydrometeor as an initial estimate: the upper limit for the x-parameter for all habits is set to 0.758, that corresponds to the pristine ice plate fully horizontally oriented. The lower limit is set to $10^{-3}$, that would correspond to a particle that is nearly spherical.

The covariance matrix for observations is defined as follows:

$$R = \frac{1}{\sigma'^2} \tag{10}$$

where $\sigma'$ is the standard error. For the dataset that has been used (e.g. Padullés et al., 2024), the $\sigma'$ can be obtained using $\sigma' = SD/\sqrt{50}$, being $SD$ the standard deviation of the differential phase obtained every 1 sec, that is, every 50 points (note that these PRO are obtained at 50Hz range).

Additionally, the standard deviations associated with the x-parameters are also calculated. Given that $J$ is the cost function, the uncertainty associated with the x-parameters is given by the covariance matrix:

$$\mathbf{C}_x = \left( \mathbf{J}^\top \mathbf{J} \right)^{-1} \tag{11}$$

Then, the standard deviation of the x-parameters is:

$$\boldsymbol{\sigma}_x = \sqrt{\mathrm{diag}(\mathbf{C}_x)} \tag{12}$$

## 2.5 ARTS look-up tables

Once the x-parameters were obtained from the optimization, we used the ARTS database to examine which particle habits correspond to these optimized values. To enable this comparison, we generated two distinct look-up tables analogous to the one presented in Padullés et al. (2025b), but based on different particle size distributions (PSD). These tables relate each particle habit in the ARTS database to its corresponding x-parameter, depending on the scattering properties of that particle. The resulting look-up tables are provided in Table A1 in the Appendix.

To improve consistency with the assumptions of the WRF microphysics schemes, two PSD formulations are implemented. First, an exponential distribution ($\mu = 0$) is used for snow, ice, and graupel in the WSM6, Goddard, and Morrison schemes, and for ice and graupel in the Thompson scheme. Second, for snow in the Thompson scheme, we adopt the PSD proposed by Field et al. (2005). Consequently, two separate look-up tables are constructed, one for each PSD configuration, allowing a more consistent linkage between WRF-derived water content and ARTS based scattering properties, while still retaining the capability of ARTS to represent nonspherical particle habits.

Finally, Figure 5 provides a schematic overview of the methodology. Starting from the WRF simulations, water contents for snow, ice, and graupel are derived under different microphysics assumptions. These are then combined with the corresponding x-parameters to compute simulated differential phase shifts ($\Delta\Phi_{sim}$). The simulated profiles are compared against the PRO observations ($\Delta\Phi_{obs}$), and the cost function $J$ is minimized to identify the optimal set of x-parameters for each microphysics scheme. In parallel, the ARTS scattering database provides precomputed x-parameters for different hydrometeor habits and PSDs. The optimized values are finally interpreted in relation to these physically based lookup tables, allowing us to evaluate which ranges of particle habits are most compatible with the observations. This workflow illustrates how the combination of WRF, ARTS, and PRO observations provides a consistent framework to assess the sensitivity of $\Delta\Phi$ to both microphysical assumptions and particle scattering properties.

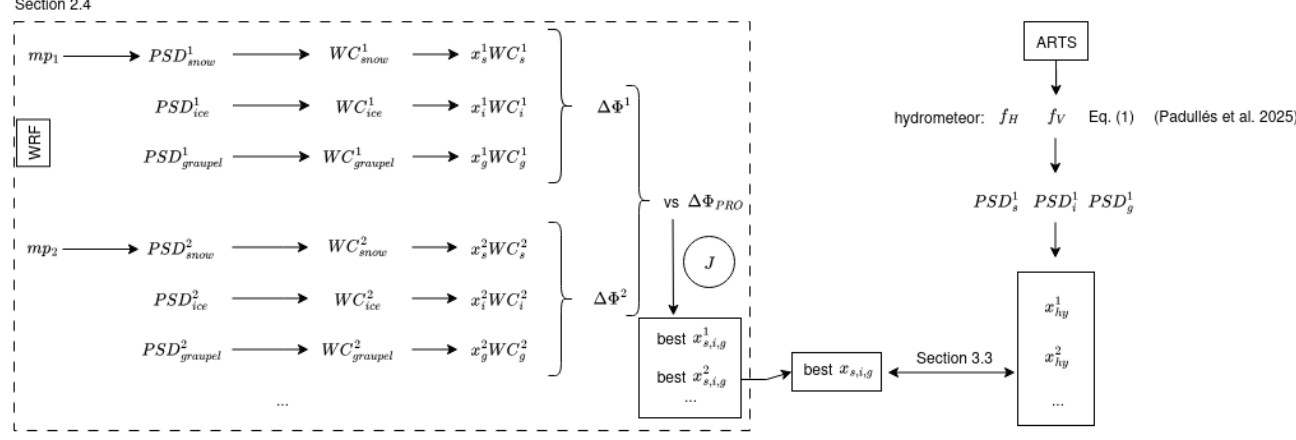

**Figure 5.** Diagram showing the steps to follow in the analysis.

## 3 Results and discussion

This section presents the results of the analysis and the corresponding discussion.

The results would inform us of which microphysics schemes in combination with different particle habits are most suitable to represent PRO observations when simulating ARs using WRF in particular, but also may provide insight to more general microphysics assumptions relevant to other modeling frameworks, including operational NWP.

### 3.1 Contribution of each hydrometeor to water content

First of all, we analyze the contribution of different hydrometeor species (i.e. rain, snow, ice and graupel), to the simulated differential phase shift, $\Delta\Phi_{sim}$. The analysis reveals that snow is the dominant contributor to the total water content, meaning that the rays travel through a larger content of snow than any other specie. A general correlation is observed between higher snow water content and an increased differential phase shift, underscoring the crucial role of snow in shaping the observed signal.

Figure 6 shows the fraction of snow WC across all ARs, for the four microphysics. With 37 cases per scheme (four simulations per AR, one for each scheme), the analysis confirms snow as the leading hydrometeor in nearly all simulations. Ice or rain typically represent the second-largest contributor, with graupel playing a smaller and more variable role. In the subsequent analysis, we will look for the best solution among the different microphysics as the one that minimizes the cost function in Equation 9.

It is also worth noting that the contributions of ice and graupel exhibit greater variability across different microphysics schemes, whereas the contribution of rain remains relatively consistent for a given AR case. In some cases, excluding the rain contribution even improved agreement with observations, but rain becomes more relevant in events with stronger low-level $\Delta\Phi$.

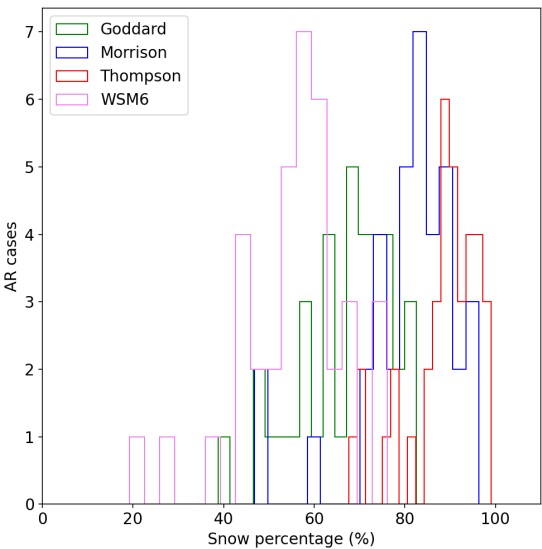

**Figure 6.** Percentage of snow water content from the total for each AR discerning between the different microphysics.

From Figure 6, it is evident that snow accounts for more than $40\%$ of the total water content in most cases. The Thompson scheme systematically predicts the largest snow WC, though this could reflect overestimation relative to others or underestimation by the other schemes, a tendency also reported at Jankov et al. (2009). Its high fraction could be linked to generally the low ice content that it generates. Meanwhile, the WSM6 presents a broader distribution, including cases where the snow water content is lower $(20\% - 40\%)$. Overall, WSM6 predicts the lowest snow percentages among the schemes, though snow remains the dominant hydrometeor.

Figure 7 shows scatter plots of vertically integrated snow water content $(\text{iWC}_{snow})$ against the integrated $\Delta\Phi$ from PRO, using values between 2 and 12 km to obtain a single representative point per case. Only the best-performing scheme (lowest cost function) is shown for each AR. In panel (a), the color indicates the optimized $x_{snow}$, while in panel (b) it shows the fraction of snow relative to the total WC. Point size reflects the cost function, with larger points meaning better agreement. The plots confirm that larger snow content generally produces stronger $\Delta\Phi$, though the relationship varies by scheme.

The scatter plots also show that the $\Delta\Phi$–snow WC relation is not universal, since values are vertically integrated and other hydrometeors also contribute. Still, a clear dependence on both $x_{snow}$ and snow fraction emerges, with a slope of about $0.015\text{mm}/(\text{kg}\cdot\text{m}^{-2})$. The fit was done without forcing the line through zero, consistent with the fact that snow is not the only driver of $\Delta\Phi$.

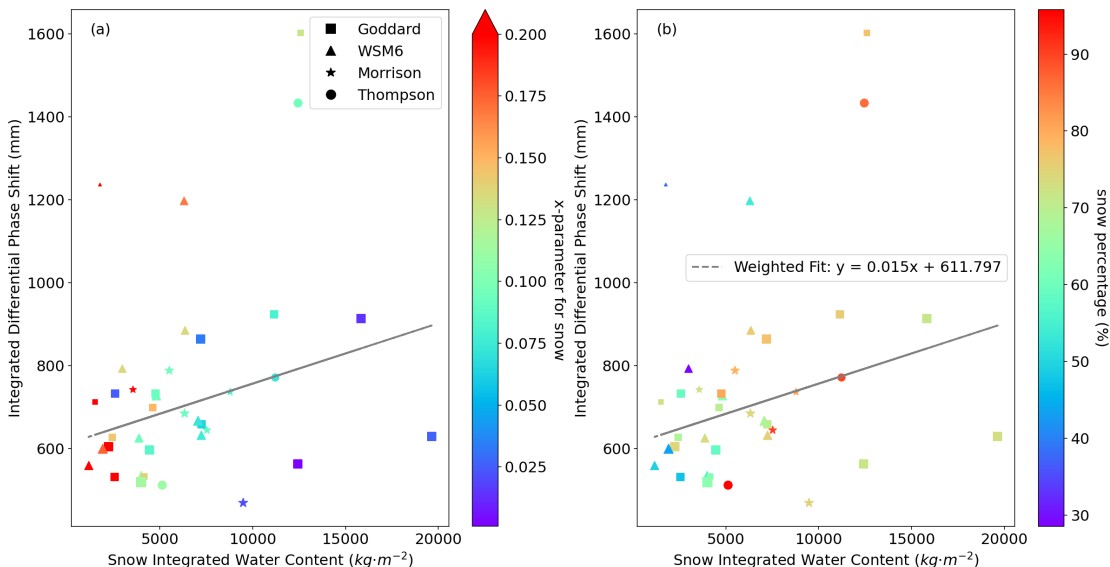

**Figure 7.** Scatter plots illustrating the relationship between integrated snow water content and integrated differential phase shift from PRO. Panel (a) presents points representing the best-performing microphysics scheme for each case, where each point is characterized by a unique marker shape corresponding to its microphysics scheme, a color scale representing the x-parameter for snow obtained through the optimization, and a size proportional to the cost function of the minimization process (larger points indicate lower costs, i.e. higher confidence). Panel (b) follows the same format but uses the color scale to represent the percentage of snow water content

To further illustrate the findings discussed above, Figure 8 displays for two examples the $\Delta\Phi$ from PRO alongside the integrated water content profiles for each hydrometeor, considering a specific microphysics scheme. It is clearly observed that snow contributes the most and closely follows the shape of the observed $\Delta\Phi$.

### 3.2 Discerning between different microphysics

The optimization process is done at two levels, within a given microphysics scheme and overall (microphysics with lowest cost function evaluated at the optimal x-parameters).This preliminary analysis aims to identify the microphysics schemes that most accurately replicate the GNSS-PRO observations.

Figure 9 presents an example of a profile $\Delta\Phi_{sim}$ obtained from the four different microphysics schemes along with the corresponding observation of PRO. In this particular case, the profile obtained using the Goddard microphysics scheme exhibits a better fit to the observed data. The x-parameters resulting from the optimization for each hydrometeor, along with the corresponding cost function value, are displayed within the same figure.

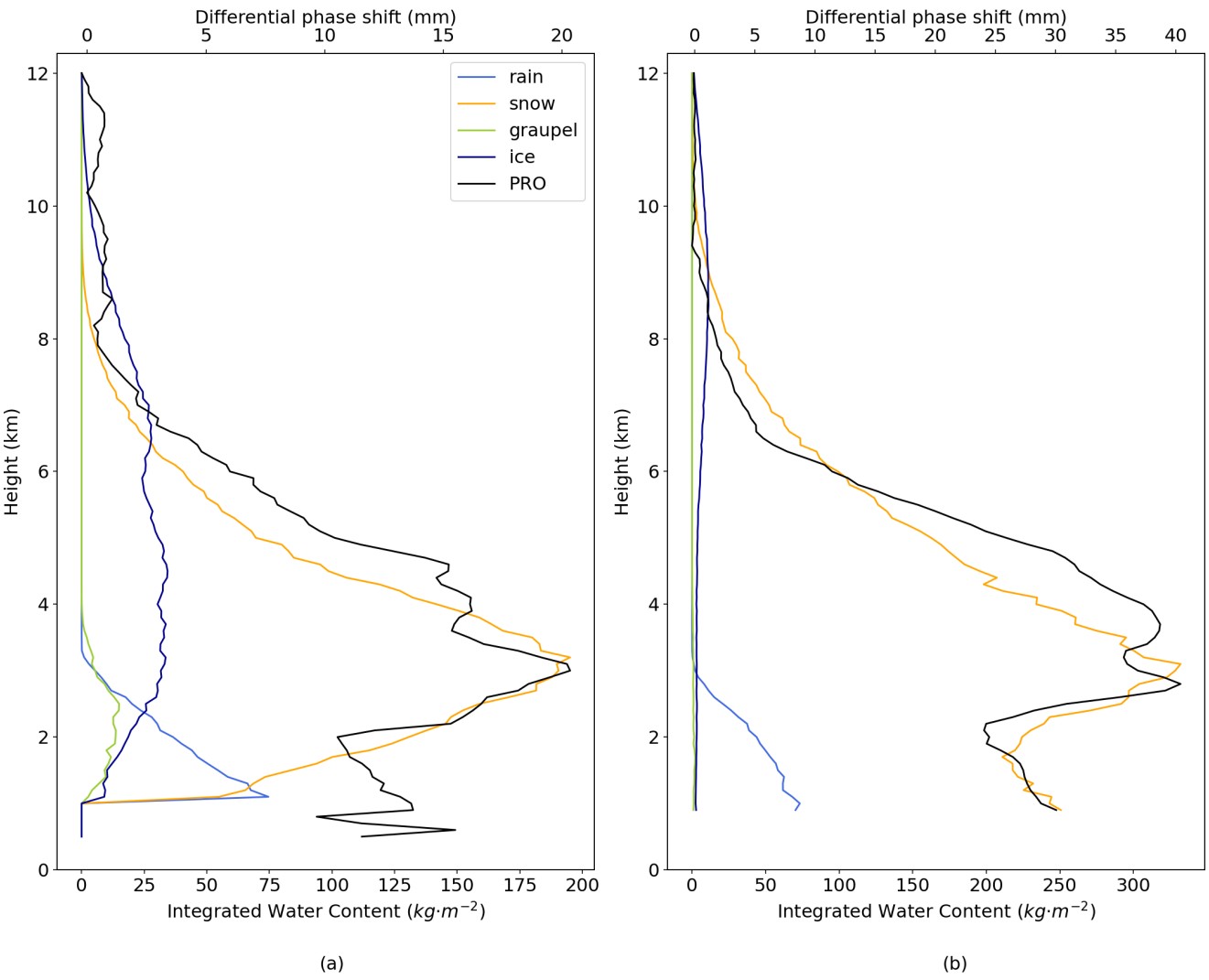

**Figure 8.** Vertical profiles of the differential phase shift (top x-axis) obtained from PAZ and the integrated water content (bottom x-axis) from WRF for the different hydrometeors. The microphysics schemes used were Goddard (a) for the observation with roid PAZ1.2019.142.10.32.G02 and Morrison (b) for the observation with roid PAZ1.2023.038.10.59.G19 scheme.

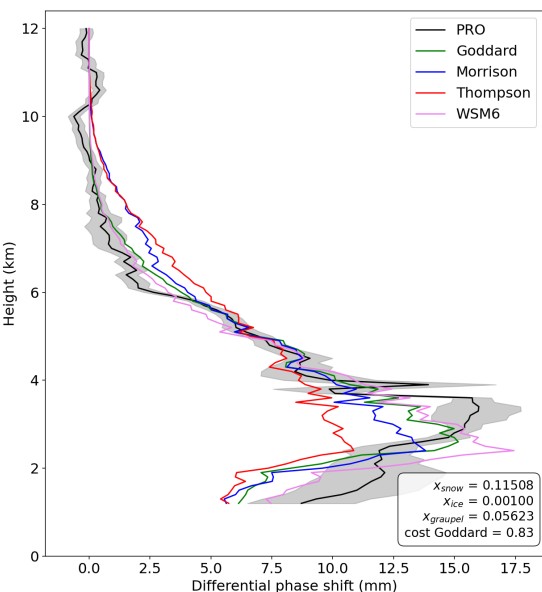

**Figure 9.** Vertical profiles of the differential phase shift obtained from GNSS-PRO and from the different microphysics of WRF. We also present the standard deviation of $\Delta\Phi$ from PRO in grey. The x-parameters optimized and the cost function associated with the best microphysics (Goddard) are presented in the legend. The roid of the observation is PAZ1.2021.034.10.07.G08.

The general results primarily depend on how each microphysics scheme simulates snow, including both its quantity and vertical distribution. Even slight variations in the snow's position can lead to differences in the results. However, the large-scale characteristics of atmospheric rivers tend to mitigate these effects, (Hotta et al., 2023).

Figure 10 presents a histogram summarizing the microphysics schemes that achieved the best agreement with PRO observa-
tions. For each case, the optimal scheme is identified based on the lowest cost function, $J$ (see Equation 9). The results indicate that the Goddard scheme consistently outperforms the others, followed by WSM6.

Further analysis shows clear differences among the microphysics schemes in how they represent frozen hydrometeors. The Thompson scheme frequently underestimates the contribution of ice to $\Delta\Phi_{sim}$, in some cases omitting it entirely, while at the same time producing the highest snow water contents compared to the other schemes, as also seen at Jankov et al. (2009). In contrast, the WSM6 scheme predicts the lowest snow water content but the highest graupel amounts, again consistent with Jankov et al. (2009), which may explain some of the mismatches with PRO observations. The Morrison and Goddard schemes differ mainly in their treatment of ice, with Morrison producing less ice that is confined to higher altitudes. For graupel, no consistent pattern emerges across the schemes. However, differences among the schemes are evident when analyzed across multiple AR cases.

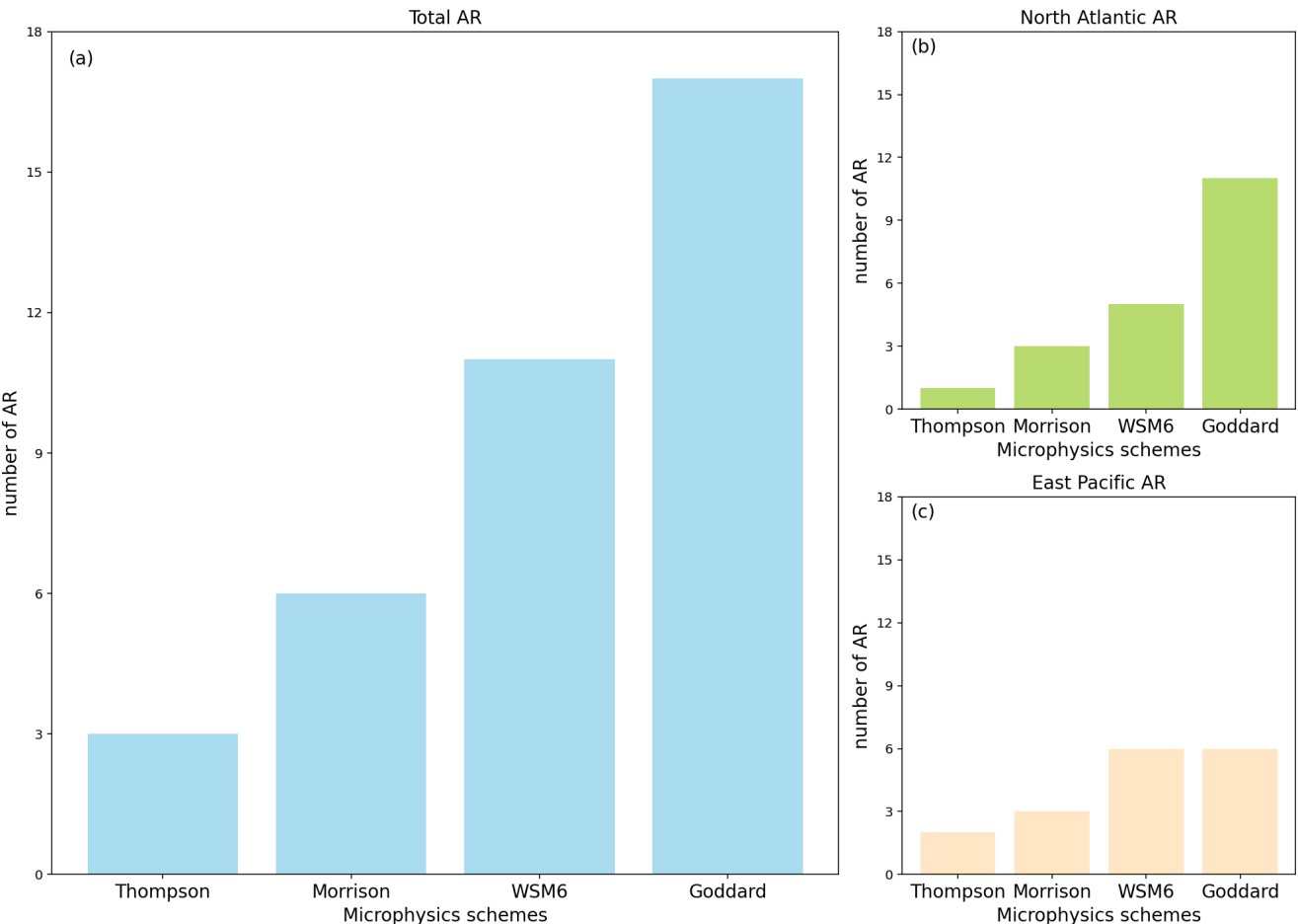

**Figure 10.** Histogram showing the number of AR that perform best with each microphysics. The performance of each scheme is evaluated using the cost function $J$ resulted from the optimization.

In terms of snow, which is the dominant hydrometeor in all schemes, no clear trends are identified across different microphysics. For rain, the contributions remain relatively consistent across all cases, suggesting that variations in performance are primarily driven by differences in the treatment of frozen hydrometeors.

### 3.3 Sensitivity to different particle habits

Building on the findings from the previous sections, namely that snow is the dominant contributor and that the Goddard microphysics scheme generally yields the best results, this section explores which particle habits provide the best agreement with observations.

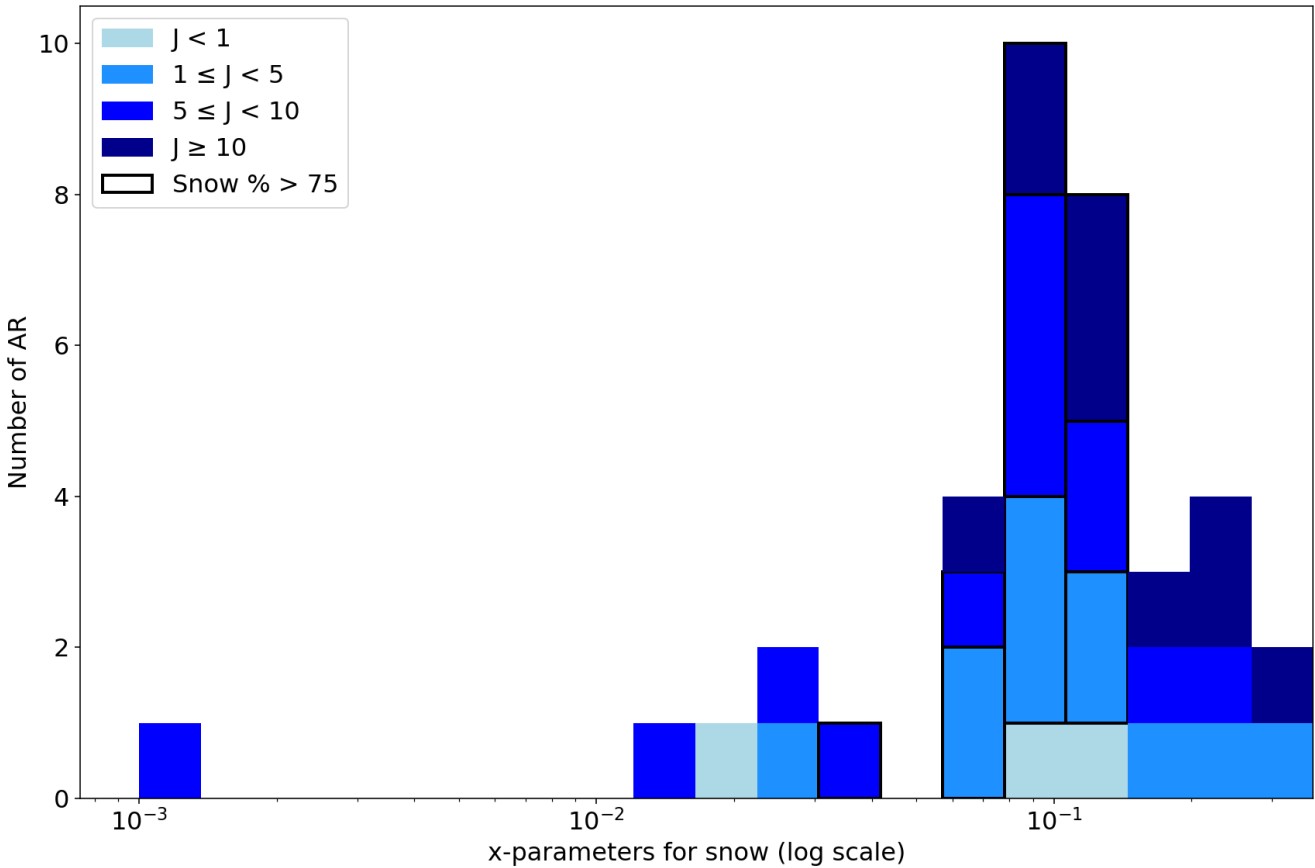

**Figure 11.** Histogram showing the x-parameters obtained from the optimization for each AR, only for snow. Darker colors show larger cost function values, while the black framing accounts for the snow WC percentage.

The parameter ranges considered during the optimization are the same for the three species that are considered here (snow, ice, and graupel), and must lay within [0.001,0.758]. These values are based on the maximum values obtained from the simulations of the particle shapes from the ARTS database, and setting $10^{-3}$ as a minimum value, equivalent to a nearly spherical habit. However, these values do not account for particle orientation, such as the distribution of canting angles. If canting information is included, habits with initially higher x-parameter may exhibit lower values, as the canting angle reduces the overall x-parameter (Padullés et al., 2025).

Figure 11 presents a histogram of the snow x-parameter values obtained from the optimization process, considering only those associated with the best performing microphysics scheme for each AR. The results indicate that the habits most closely aligning with observations are those with x-parameter values around $0.05 - 0.15 mm/(kg m^{-2})$, which correspond to particles such as bullet rosettes and snow aggregates, as we can see from Table A1.

In the same Figure 11, we distinguish cases where the snow water content percentage exceeds 75%. As shown, these cases are associated with $x_{snow}$ values again ranging approximately from 0.05 to $0.15 mm/(kg \, m^{-2})$. In addition, we differentiate cases based on the value of the cost function.

The predominance of snow aggregates and bullet rosettes as the optimal habit aligns with physical expectations, as these particles exhibit complex and irregular structures. The loosely bound individual crystals within aggregates influence their scattering properties. In contrast, cases where snow is classified as plate-like structures show a significant overestimation of snow contribution. This discrepancy likely arises from the highly directional scattering behavior of snow plates, which results from their compact and well-defined crystalline structures, leading to an overestimation in observed values. These findings underscore the importance of accurate microphysical parameterization in numerical models to properly represent snowfall characteristics.

Similarly, histograms analogous to Figure 11 were generated for graupel and ice (not shown). In general, the x-parameters for ice and graupel are much lower than those for snow, consistent with snow being the main contributor to $\Delta\Phi$. This effect is particularly evident for graupel, where most cases show values very close to the lower limit. In contrast, ice, although also showing a prominent peak near this limit, exhibits a somewhat broader distribution across the entire range of x-parameter values.

Figure 12 presents the summary of the values of the x-parameters corresponding to the best-performing microphysics scheme for each AR, for snow, ice, and graupel. The first relevant thing to note is that the optimized values for ice and graupel often lay over the lower limit (the allowed range of x-parameters values is shown with a gray shaded area). Furthermore, the large error bars for these species indicate that they have a minor contribution to $\Delta\Phi$, since the minimization does not depend much on their value. Overall, the smallest error bars are obtained for snow, and its x-parameter values indeed appear to follow a more consistent trend around approximately $x = 0.1 mm/kg \, m^{-2}$ (as seen in Figure 11). Additionally, Figure B1 in the Appendix presents a 2D plot of the cost function as a function of the x-parameters for snow and ice. Each plot corresponds to a specific observation, with the case PAZ1.2018.234.06.45.G06, Figure B1 (a), representing one where the error bars in Figure 12 are relatively small, and PAZ1.2022.122.01.17.G25, Figure B1, representing a case with substantially larger error bars. In both plots, the x-parameter for graupel is set to the minimum value obtained for each respective case. This figure illustrates the distribution of the cost function for one scenario in which the parameter values significantly influence the final agreement, compared to a case where their relevance is diminished. Notably, the contours for PAZ1.2018.234.06.45.G06 appear more defined around specific x-parameter values, whereas the contours in the other case are more diffuse, indicating less sensitivity to parameter selection.

Additionally, we also present a figure in the Appendix showing the best x-parameters for snow as a function of the microphysics scheme (Figure C1). In general, we observe that the same trend holds across all microphysics schemes, as in Figure 11 most of the cases have x-parameter values around $0.05 - 0.15 \, mm/kg \cdot m^{-2}$.

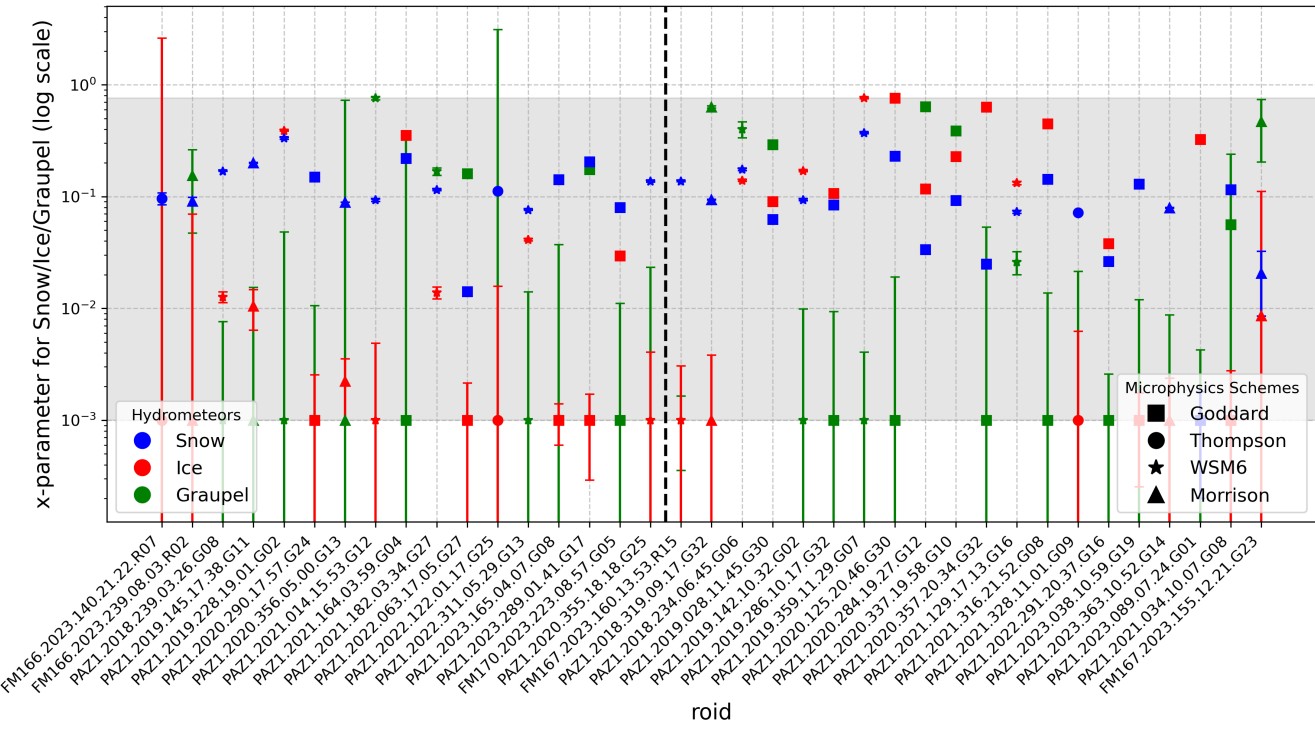

**Figure 12.** Scatter plot showing the different values of the x-parameters for the best-performing microphysics scheme for each AR case. Additionally, error bars are presented for each value, along with the optimization range of the hydrometeors (shaded colored area). The y-axis is in logarithmic scale. The dashed line represents a division between the regions of each observation: those on the left correspond to the Pacific, while those on the right correspond to the Atlantic.

## 3.4 ARs over the North Atlantic and East Pacific

Given that ARs occurring in the Atlantic and Pacific originate from distinct moisture sources and are exposed to varying atmospheric conditions, it can be hypothesized that there might be differences in the vertical distribution of hydrometeors.

In this section, after analyzing the different cases and reviewing the results from the analysis already discussed, we will examine whether the location of the AR influences the conclusions we can draw. In total, we have 37 ARs, with 20 located in the north Atlantic and 17 in the north-east Pacific.

From what concerns to the microphysics schemes, Figure 10 (b) and (c) shows the Goddard scheme is the most frequently identified as the best performing scheme across all AR, suggesting that it provides the most consistent agreement with PRO. The WSM6 scheme also performs well but is less dominant than Goddard. The Morrison and Thompson schemes appear less frequently, indicating that they may not be as effective in reproducing the observed conditions. It seems that the general trend observed for the 37 ARs is the same as in the individual cases of the Atlantic and the Pacific, but for the Pacific cases we can highlight the same good performance of Goddard and WSM6, fact that is not as remarkable as in the Atlantic cases.

Another distinction emerges in Figure 12 when comparing Atlantic and Pacific AR. As shown by the dashed line in the same figure, which separates Pacific (left) from Atlantic (right) AR. The particle habits for ice in the Atlantic AR tend to have higher x-parameters compared to those from the Pacific, suggesting that ice is given by particles with higher scattering in the Atlantic cases. Also, the error bars for the ice are larger for the Pacific cases compared to the Atlantic ones. Above all, for graupel, same happens for the error bars in comparison with the Atlantic. This suggests that in certain Pacific cases, the selection of the particle for graupel and ice does not play an important role.

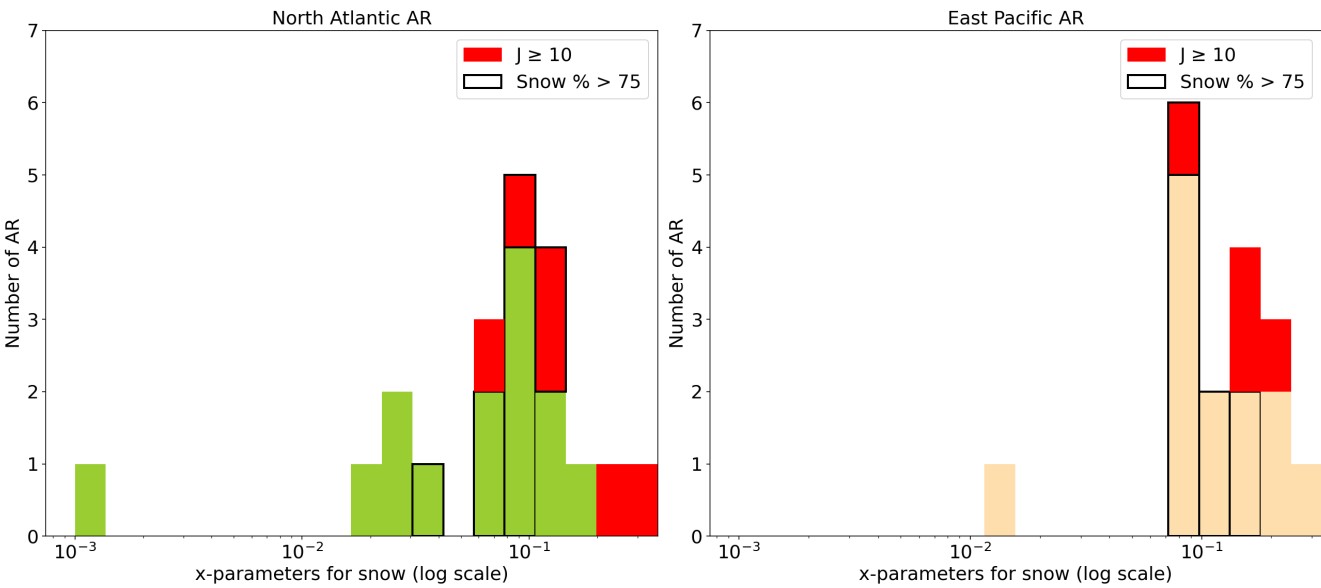

**Figure 13.** Histograms showing the best x-parameters for the different AR as sin Figure 11 but discerning between those AR from the Pacific (a) and from the Atlantic (b).

Regarding the type of particle habits, we present the following histograms in Figure 13. From this Figure 13, both panels show same general trend for the snow x-parameters, however for the East Pacific cases the values are more concentrated between the $0.05 - 0.15 mm/kg \cdot m^{-2}$ range.

## 4 Conclusions

This study demonstrates that GNSS-PRO is an effective tool for distinguishing between different microphysics parameterizations and various particle habits in the modeling of ARs. The results obtained from WRF and ARTS simulations show strong agreement with PRO observations, particularly in capturing the differential phase shift, $\Delta\Phi_{sim}$. Among the hydrometeors considered, snow emerges as the dominant contributor to $\Delta\Phi_{sim}$, underscoring its critical role in shaping the observed signal.

The Goddard microphysics scheme consistently provides the best performance across cases, likely due to its more balanced representation of hydrometeors.

The effectiveness of each microphysics scheme varies depending on the specific AR analyzed; however, certain trends remain consistent. The Thompson scheme tends to generate lower ice concentrations while overestimating snow water content, which negatively impacts its agreement with GNSS-PRO. Conversely, the Goddard and WSM6 schemes yield the highest ice

concentrations, while the Morrison scheme generally predicts lower ice content. These differences highlight the significant influence of microphysics parameterizations on model accuracy and their implications for interpreting observational datasets. The incorporation of GNSS-PRO observations into microphysics development could enhance the refinement of parameterizations, particularly for snow and other frozen hydrometeors.

Furthermore, while ice and graupel exhibit varying contributions depending on the microphysics scheme, the role of rain

remains relatively consistent across different schemes for the same AR. In most cases, including or excluding rain does not significantly impact the agreement between $\Delta\Phi_{sim}$ and $\Delta\Phi_{obs}$, except when the maximum $\Delta\Phi$ occurs at lower altitudes closer to the surface. This behavior aligns with expectations, as rain is a more spatially variable phenomenon compared to frozen hydrometeors.

Regarding the x-parameter for snow we have seen that even though there is variability among cases the results tend to

435 systematically group around 0.05-0.2 $mm/kg \cdot m^{-2}$. This corresponds to habits like aggregates, columns, and bullet rosettes, and discards the systematic presence of horizontally oriented plate-like habits. Likewise, the presence of orientable shaped habits is required in order to explain the observations, and therefore we can discard snow being contributed by spheres or habits that are totally randomly oriented.

The analysis also reveals some differences between Atlantic and Pacific ARs in terms of particle habits, while microphysics

schemes do not seem to play a significant role in distinguishing between them. Interestingly, for Atlantic ARs, the best-performing ice habits tend to be more distributed than those in Pacific ARs, suggesting a greater relative contribution of ice in these cases.

It is important to note that the densities, and shapes assumed in the WRF microphysics schemes are not fully consistent with those represented in the ARTS scattering database. However, both sources are consistent in certain aspects, like the

445 PSDs. We do not consider a full consistency to be significat because our aim is different: we take water content directly from WRF and computed the x-parameters that suit best the observations to then assess which nonspherical particles from ARTS are most compatible with them. A fully consistent treatment of WRF assumptions would require the use of spherical particles, which would yield no $\Delta\Phi$ and thus no meaningful constraint from PRO. Instead our approach exploits the sensitivity of PRO to nonspherical hydrometeors, providing insight into which microphysical representations of water content are more

compatible with satellite observations. We therefore acknowledge this trade-off and consider the present framework a first step, while noting that future work could refine the consistency between WRF and ARTS. However several studies have already demonstrated that the coupling of both is succesful when representing satellite observations (e.g. (Wang et al., 2016)).

Overall, these findings underscore the importance of selecting appropriate microphysics schemes to accurately represent snow processes and the necessity for further research into the sensitivity of these schemes to observational and modeling

uncertainties. The results also highlight the need for additional validation using independent datasets to improve the reliability of hydrometeor property retrievals.

As for future work, it would be interesting to consider additional aspects of particle habits when analyzing the agreement with PRO observations. Factors such as the canting angle or the use of different habits for the same hydrometeor specie (i.e. snow, ice, graupel) depending on variables like temperature could provide further insights.

*Data availability.* The WRF code was downloaded from https://github.com/wrf-model. The PAZ PRO observations (10.20350/digitalC-SIC/16137) are obtained from https://paz.ice.csic.es/dataAcces.php. The Spire PRO observations have been obtained through the ESA Earth Observation (EO) Third Party Mission (TPM) program contract Id. PP0098939.

*Author contributions.* AP, RC and EC have planned the conceptualization of the analysis; AP and RP have done the methodology; AP has developed the software; AP and RP have performed the data analysis; AP has wrote the manuscript draft; RP and EC have reviewed and 465 edited the manuscript.

*Competing interests.* The authors declare no competing interests

*Acknowledgements.* This publication is part of the Grants RYC2021-033309-I and, PID2021-126436OB-C22 and PID2024-155592OB-C22 funded by the MCIN/AEI (10.13039/501100011033) and the European Union "NextGenerationEU/PRTR" and "ERDF A way of making Europe". Work performed at the ICE-CSIC was also partially supported by the program Unidad de Excelencia María de Maeztu CEX2020-470 001058-M. Part of the investigations at ICE-CSIC, IEEC are done under the EUMETSAT ROM SAF CDOP4. Access to Spire PRO data has been granted through ESA Third Party Mission (TPM) porposal PP0098939.

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

## Appendix A: Appendix1

Table A1 shows the different values we obtained for the $K_{dp}/WC$ relationship for the different ARTS particles, according to the PSD used. In addition, we also present the values obtained in Padullés et al. (2025).

## Appendix B: Appendix2

Figure B1 complements the earlier Figure 12 by presenting a two-dimensional sensitivity map of the cost function with respect to the x-parameters of snow and ice, where the graupel values are fixed. This visualization helps us understand the interdependency between these two hydrometeors in terms of their influence on the differential phase shift.

Figure B1 (a), shows an observation (PAZ1.2018.234.06.45.G06) corresponding to an optimization result with small error bars, indicating high confidence in the measured differential phase shift. The cost function map for this case is more structured, with a well-defined minimum region. This suggests that the optimization process is better constrained, and that we can identify

| particle | x-parameter (Padullés et al. 2025) | x-parameter (Exponential PSD) | x-parameter (Field et al. 2005) |
|---|---|---|---|
| HongPlate_Id9 | 0.758 | 0.545 | 0.750 |
| LiuThinPlate_Id16 | 0.650 | 0.619 | 0.649 |
| HongColumn_Id7 | 0.556 | 0.317 | 0.547 |
| IconCloudIce_Id27 | 0.606 | 0.459 | 0.598 |
| LiuLongColumn_Id14 | 0.425 | 0.433 | 0.425 |
| LiuSectorSnowflake_Id3 | 0.666 | 0.268 | 0.630 |
| LiuThickPlate_Id15 | 0.388 | 0.388 | 0.388 |
| LiuShortColumn_Id13 | 0.269 | 0.268 | 0.269 |
| HongBulletRosette_Id5 | 0.273 | 0.195 | 0.266 |
| EvansSnowAgg_Id1 | 0.051 | 0.313 | 0.076 |
| HongBulletRosette_Id11 | 0.272 | 0.181 | 0.257 |
| HongBulletRosette_Id10 | 0.160 | 0.113 | 0.162 |
| HongAggregate_Id8 | 0.096 | 0.097 | 0.096 |
| HexColAggCrystal_Id22 | 0.025 | 0.138 | 0.027 |
| HexPlaAggCrystal_Id20 | 0.027 | 0.278 | 0.027 |
| HexColAggCrystal_Id18 | 0.017 | 0.197 | 0.019 |
| LiuBlockColumn_Id12 | 0.072 | 0.078 | 0.072 |
| HongBulletRosette_Id2 | 0.086 | 0.046 | 0.079 |
| IconSnow_Id28 | 0.028 | 0.124 | 0.034 |
| HexPlaAggCrystal_Id19 | 0.017 | 0.119 | 0.017 |
| GemSnow_Id32 | 0.008 | 0.103 | 0.009 |
| HexColAggCrystal_Id21 | 0.017 | 0.080 | 0.017 |
| HexColAggCrystal_Id17 | 0.010 | 0.088 | 0.011 |
| HongBulletRosette_Id4 | 0.016 | 0.027 | 0.020 |
| HongBulletRosette_Id6 | 0.012 | 0.015 | 0.012 |
| Rosette_Id36 | 0.017 | 0.011 | 0.018 |
| GemCloudIce_Id31 | 0.019 | 0.018 | 0.006 |
| GemHail_Id34 | 0.003 | 0.046 | 0.002 |
| IconHail_Id30 | 0.008 | 0.050 | 0.005 |

**Table A1.** Table showing the different values of the $K_{dp}/WC$ relation for the different ARTS's particles.

a relatively precise combination of x-parameters for snow and ice that provides the best match to the observed data. The
contours guide us to this optimal region, typically centered around x-parameters for snow close to $\sim 0.1 mm/kg \cdot m^{-2}$.

In contrast, Figure B1 (b) shows an observation (PAZ1.2022.122.01.17.G25) with very large error bars, where the uncertainty
in the measurement significantly reduces our ability to constrain the optimal solution. The cost function is relatively flat across
a broader region, with the minimum being less defined and spread over a larger range of x-parameters. Multiple combinations
of snow and ice x-parameters yield similar low cost values, making it harder to select a dominant particle habit.

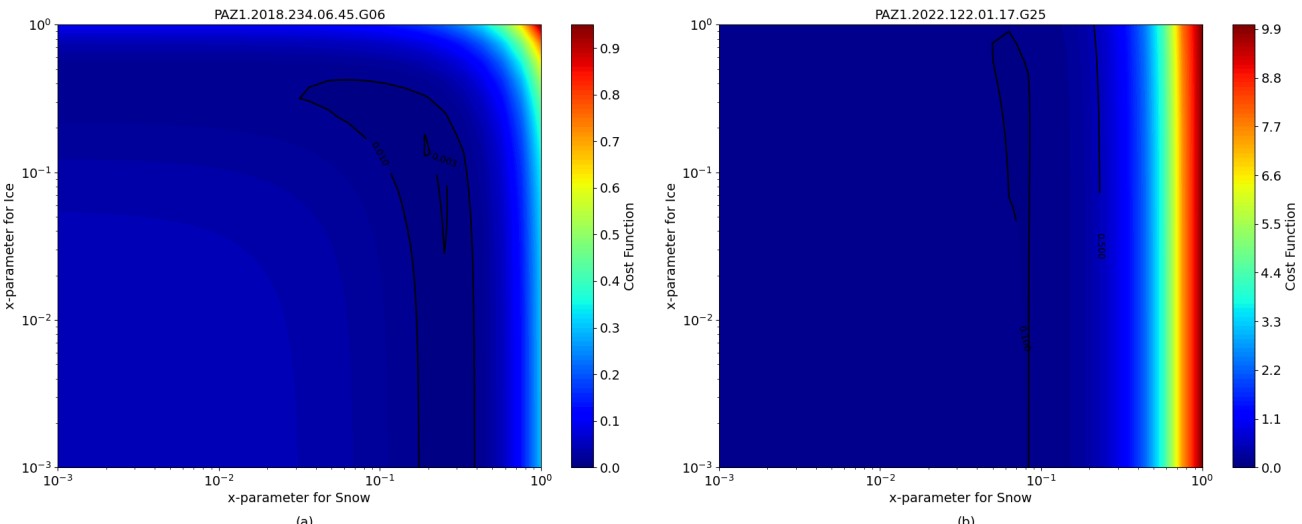

**Figure B1.** Two-dimensional plot showing the cost function values associated with two cases depending on the values of the x-parameters
for snow and ice. The x-parameter values associated with graupel are constant and correspond to the minimum value.

## Appendix C: Appendix3

From the analysis of particle habits, we also present Figure C1, which shows the optimal x-parameters for snow depending on
the microphysics.

Notably, for the Thompson microphysics scheme, most ARs exhibit a snow water content percentage above $75\%$, as indicated
by the black border bars. This suggests that the Thompson scheme systematically simulates a dominance of snow in these
events. In this scheme, we can also see a broader presence of higher cost function values, possibly indicating challenges
in fitting PRO observations despite high snow representation. For Goddard and WSM6, both show a group of optimal x-
parameters around the $0 - 0.2 mm/kg \cdot m^{-2}$ range, again pointing to aggregates and bullet rosettes as the most representative
snow habits. Whereas for Morrison, we still observe a concentration in this same range, however we can distinguish a peak
around $0.1 mm/kg \cdot m^{-2}$.

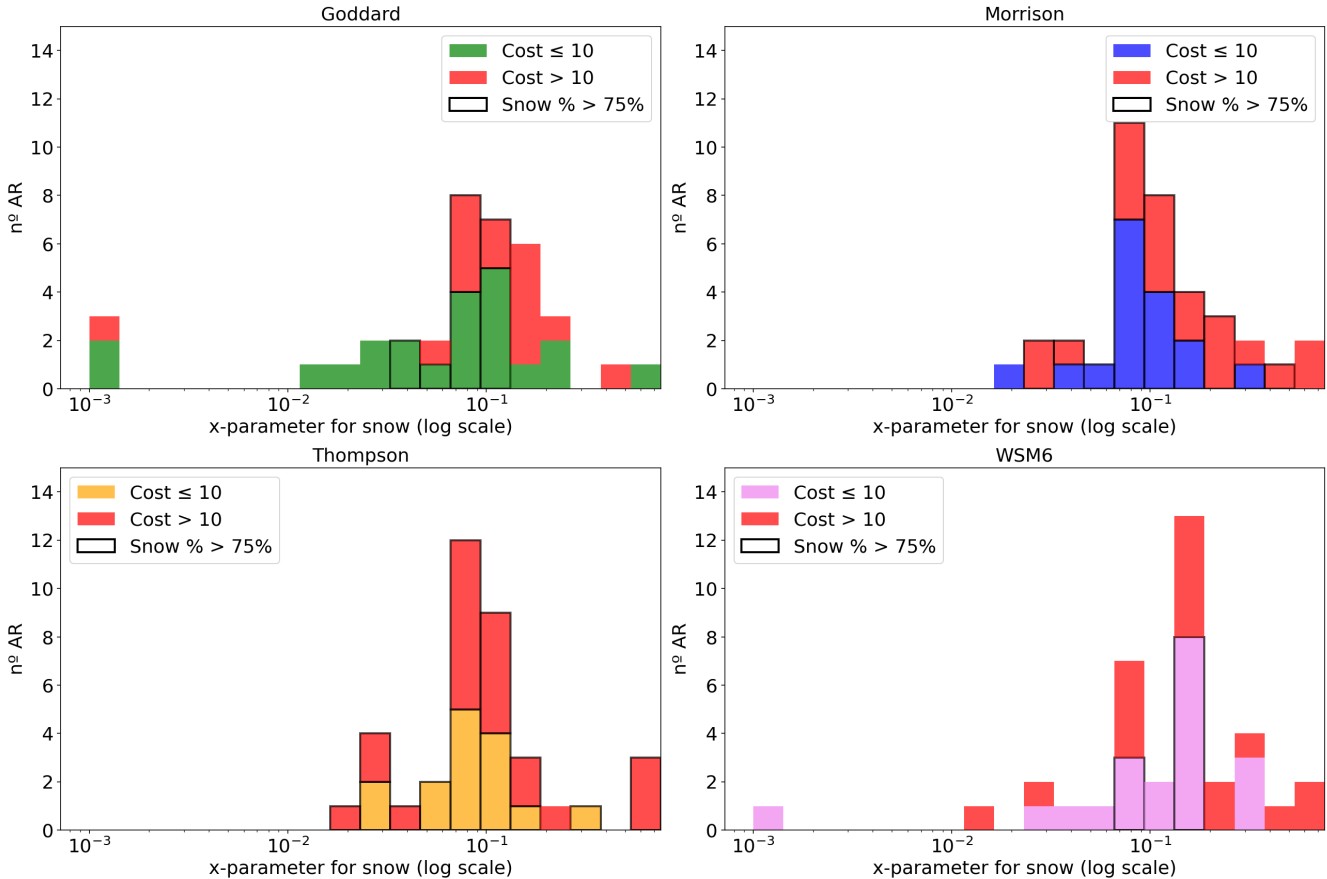

**Figure C1.** Histograms showing the particles for snow resulted from the optimization process differentiating between the microphysics schemes.

Overall, this figure supports the idea that aggregates and bullet rosettes consistently yield better agreement with PRO, and that x-parameters around $0.1 mm/kg{\cdot}m^{-2}$ are particularly effective across various schemes. It also highlights how the microphysics schemes differ in their hydrometeor composition trends, and therefore in their accuracy for representing PRO observations.