# Peer review of "Constraining microphysics assumptions on the modeling of Atmospheric Rivers using GNSS Polarimetric Radio Occultations"

_EGUsphere, 2025_

## Referee Comment (RC1)

Egusphere-2025-1950.pdf

Constraining microphysics assumptions on the modeling of
Atmospheric Rivers using GNSS Polarimetric Radio Occultations

Authors: Antía Paz, Ramon Padullés, and Estel Cardellach

**General**

The manuscript is novel and reports on unique way to assess the representation of hydrometeor habits in numerical weather prediction models.

Abstract and introduction. These sections start off with PRO technical details rather than "why" you are doing this work in the first place. Give a reason, some rationale. Suggest something like this:

Improving the prediction of precipitation intensity remains an elusive goal for the operational weather community. In current cloud resolving forecast models, the introduction of hydrometeors and the associated latent heat release are represented with either convective parameterizations or microphysical parameterizations or both, depending on the resolution of the simulation (Hristova Veleva et al 2021, https://doi.org/10.3390/atmos12020154). This study aims to assess the use of PRO in constraining the choice of microphysical assumptions within models, by exploiting the sensitivity of the PRO technique to the model's forecasted water content and selection of hydrometeor shapes.

Section 2.3 is confusing as written with the "x-parameter" terminology. It's a strange sort of ad-hoc choice of terminology for a quantity that is fundamental to the material in the remainder of the article. If I understand your discussion, this is the "A" quantity in Eqn (6).

**Specific**

Near Line 30. Suggest this wording: GNSS systems such as GPS transmit in a Right Hand Circularly Polarized (RHCP) state. A PRO-capable RO receiver uses dual orthogonal receive polarizations, horizontal (H) and vertical (V), enabling the measurement of differential phase delay ($\Delta\phi$, defined in Section 2.1 below). This differential delay is induced when the transmitted GNSS signals propagate through nonspherical hydrometeors (such as raindrops) in the atmosphere (Cardellach et al., 2015).

Line 54. The work by Murphy et al 2019 was the original work that proposed investigating cloud resolving model microphysics. Also your work by Shu-Ya Chen fpt TC cases predates your study https://egusphere.copernicus.org/preprints/2025/egusphere-2024-3708/).

Suggest this wording: Murphy et al (2019) proposed using PRO simulations to examine the model sensitivity to assumptions in its microphysical parameterization. However, PRO data were not available at that time. A recent study by Chen et al (2025, in review) compared observed and simulated ROHP data to examine a small number of tropical cyclones (TC) events. Our work focuses on the much larger number of PRO data from ROHP and Spire that cover Atmospheric River (AR) weather events. Together with PRO simulations, our comparisons aid in determining the extent to which variations in microphysical schemes in AR's can be discerned with PRO observations.

Line 58: However, the microphysical schemes implemented in mesoscale models such as WRF do not explicitly provide the required scattering characteristics at the GNSS frequencies (near 1.4 GHz) and limb-viewing angles needed to perform the Kdp computation.

Line 66. You need to cite the Hotta et al (2024) paper here, it was the first paper to simulate and study ROHP data in several AR cases. As they concluded, the AR cases provided the best agreement with ROHP owing to the nature of the weather event (more widespread, hence less variability along the long ray paths, relative to the TC cases with more variability from convective conditions). This is another reason why AR are good choices of weather events to examine.

Near Line 95. Most non-RO people will be unfamiliar with the term "ray path". The word is a construct used to refer to the volume of air that is sampled (discretely in time) by the PRO receiver as the RO rises or sets. I think a cross section figure would be helpful to explain what the path length is. You can provide one or refer to several in the Padulles et al papers, or cite the Figure from the summary paper from the "2023 Polarimetric Radio Occultations Workshop" in BAMS. https://doi.org/10.1175/BAMS-D-24-0050.1

Figure 1. The rays are from which PRO data? Perhaps make this for the case shown in Figure 7 so the reader has an idea of which rays sliced through the storm and where.

Figure 4 caption. Be more specific, i.e., "Longwave IR brightness temperature image from geostationary satellite data".

Line 180. The term "x-parameters" is confusing. So, the x-parameters are the A and B terms in Eqn (6), for each level of the PRO profile, for each specie? (those relate Kdp and WC). Or are you fixing B for each microphysics type and letting only "A" be the "x-parameter". Explain this better as you refer to "x" a lot from this point on and is a source of confusion.

Line 214. Equation 8. One is used to seeing this type of equation for a data assimilation scheme in a forecast model. In that case, one can envision the "initial estimate" (the model background state). What is the background state (Xb) here, and how is it obtained? While y and H(x) are easy to envision- H(x) is your simulated $\Delta\phi$ and y is the observed $\Delta\phi$.

Line 225. From my understanding, you are assuming a diagonal background covariance matrix with all off-diagonal terms set to zero. Is correct, and if so, is this realistic? No need to change, just explain and justify.

Figure 6. It's a stretch to see much of a relation here. Why not plot the integrated $\Delta\phi$ above, say, the -10 C level (where the bulk of the ice is) instead? If there is a relation, it may be better revealed.

Line 282. You say, "the minimization of J is performed jointly for both the microphysics scheme and x-parameter". Should not this have been mentioned back when you introduced Eqn (8)?

Line 292. You say, "Even slight variations in the snow's position can lead to differences in the results. However, the large-scale characteristics of atmospheric rivers tend to mitigate these effects." For the first point, do you think that this is important, even for AR events? Cite the Hotta et al. 2024 manuscript for this last point.

---

## Referee Comment (RC2)

**Review comments for *Constraining microphysics assumptions on the modeling of Atmospheric Rivers using GNSS Polarimetric Radio Occultations**

This paper investigates the potential of Polarimetric Radio Occultation (PRO) to provide insight into hydrometeor vertical structures, using WRF simulations and the ARTS scattering database to simulate differential phase ($\Delta\Phi$) for comparison with satellite observations. The use of PRO in this context is still emerging, and this study contributes to ongoing efforts to assess its sensitivity to different microphysical assumptions. Nonetheless, continued exploration of PRO for hydrometeor evaluation is valuable, and further development of physically consistent frameworks is encouraged.

That said, a fundamental concern arises from the physical inconsistency between the WRF microphysics (MP) schemes and the scattering properties derived from the ARTS habit database, which fundamentally undermines the interpretation and validity of the results.

1. Mismatch Between WRF MP Assumptions and ARTS Scattering Properties

The WRF MP schemes (e.g., Thompson, Morrison) are bulk microphysics parameterizations. They assume a fixed set of physical properties for each hydrometeor category: shape, density, and particle size distribution (PSD). For example, snow in the Thompson scheme may be represented by soft aggregates with a specific mass-size and PSD relationship, while graupel has a different assumed density and terminal velocity. These assumptions are not explicitly output but are embedded in the diagnostic formulas that compute water content and number concentration.

In contrast, the ARTS habit database provides scattering properties (x-parameters) computed for discrete particle habits, each associated with its own shape, refractive index, and PSD assumptions. These may include bullet rosettes, dendrites, plates, spheres, or irregular shapes, and often with a PSD that differs from the one assumed in WRF MPs.

By using x-parameters from ARTS in combination with WC fields from WRF, the method effectively combines microphysical representations that were never meant to work together. This is a physical inconsistency — mixing the absorption/extinction/scattering characteristics of one assumed hydrometeor population with the mass distributions of another. Even if the x-parameters are bounded within ARTS values, this does not correct the mismatch in underlying particle physics.

2. Optimization Does Not Evaluate Microphysics Skill

The second issue concerns the interpretation of the optimization procedure. The authors optimize the x-parameters (within ARTS bounds) to minimize the difference between observed and

simulated differential phase shift (Kdp), holding the WRF-derived WC fixed. This means that any biases or errors in the WC fields are effectively absorbed by tuning the x-parameters.

Therefore, the optimization outcome does not evaluate whether the MP scheme is accurately predicting WC or phase shift — it only shows which x-parameters (within ARTS-defined bounds) can reconcile the WC fields with observations. This decouples the validation from the actual physical outputs of the MP scheme. Two different MP schemes could produce very different WC fields, but the optimization could find different x-parameters for each that yield similarly good fits to observations — misleadingly suggesting both are good, or that one is better based on fitted x alone.

3. Misuse of "Forward Operator" Terminology

The paper also uses the term "forward operator," which typically refers (in data assimilation) to a physically consistent transformation of model state variables (like WC) into observation space (like Kdp), based on known physics. In a proper forward operator, the mapping is fixed, and the state variables are adjusted (via a cost function) to minimize differences from observations.

However, in the present methodology, the forward operator is not fixed — it is being modified by optimizing x-parameters. This blurs the boundary between model physics and observation operator and makes it unclear what is actually being evaluated. Optimized x-parameters are not state variables and cannot be used to adjust the model state or improve forecasts, thus limiting the method's value even in a diagnostic context.

4. Suggested Alternative

A more physically meaningful approach would be to:

(i) Use x-parameters consistent with each MP scheme's assumed particle properties, either by matching ARTS habits or computing scattering from scratch;

(ii) Use those fixed x-parameters to compute simulated Kdp, and compare directly to observations, without optimization;

(iii) Evaluate the WC fields directly by assessing how well they reproduce observed Kdp under physically consistent scattering assumptions.

Summary

In short, while the idea of using observations of phase shift to evaluate model microphysics is important and timely, the current implementation is flawed due to the physically inconsistent blending of ARTS scattering properties with MP-derived WC, and an optimization procedure that does not test the microphysics predictions directly. The conclusions drawn about MP scheme performance are therefore not supported by the methodology, and a revision of the experimental design is recommended.

Besides, there are some other comments related as follows:

L43: Passive microwave radiometers have also been utilized to interpret precipitation vertical structures, (Turk…

Comma here should be removed

L47: These change depending…

These changes depend

L49: simulations will be conducted…

were conducted

L100: As the PRO rays traverse the derived from PRO.

This is a broken sentence

L101: provides valuable insightse from GPS to LEO

Typo "insightse"

L101: This, in turn, provides valuable insightse from GPS to LEO, refractivity gradients cause bending, resulting in rays becoming tangent to the surface at their lowest point, termed the tangent point, ht.

This whole sentence is not grammarly correct. Please fix.

L107: matching a regular grid between 0 and 20 km

What was the reason to pick top height at 20km? Is this limit of the WRF model top?

L127: For some of the simulations, instead of using the RRTMG schemes for shortwave and longwave radiation, the New Goddard Shortwave and Longwave Schemes were used due to certain errors that occurred in specific simulations

Could these mis-match lead to any difference among MP results used in the following comparison?

L139 whether hail or graupel is the third class of ice however,

A comma is needed before "however"

Section 2.2.1: The numbering of this section seems weird since there is no other subsection in section 2.2.

L215: where the x_b comes from? Is it from ARTS?

If so, please refer to my general comment

L266: figure captions (a) and (b) are opposite to with what are defined in Figure 6. Caption

L282: "Even though the minimization of J is performed jointly for both the microphysics scheme and x-parameter"

According to L210, x-parameters are obtained by optimizing the cost function as in (8) by fixing WC. Therefore, I don't understand what you meant as jointly.

L293: The results indicate that the Goddard scheme consistently outperforms the others, followed by WSM6.

Since the simulation has used optimized x-parameters, which is case dependent (I think if there are other 32 points, you would get a different set of x-parameters), it is hard to say which original WRF simulation, and its corresponding scheme are the best. We can only say the Godard scheme was compensated by the specific "optimized" x_parameter in this particular case and such a combination performs the best.

Line 333 The first relevant thing to note is that the optimized values for ice and graupel often lay over the lower limit (the allowed range of x-parameters values is shown with a gray shaded area and related discussions

This suggests the optimization problem is under-determined and constrained in a way that prevents a proper minimization of the cost function. As a result, I would question how robust the resulting conclusions are regarding the relative contributions of different hydrometeor species— especially if the optimization is not genuinely being achieved for some of them. The physical reason for not being able to minimize some of the MP results is that these MP schemes use different assumptions of the hydrometeor habits and therefore, you might not be able to get a convergence at all by using ARTS defined habits.

In its current form, the method raises a fundamental question about physical inconsistency between ARTS habits and MP assumptions. It might be helpful to examine whether the achieved x-parameters through Eq (8) match the assumption of the specific MP. If so, it might be meaningful to give some insights on habit distribution, matching the bulk parametrization but not described explicitly in MP. But it seems not the scope of current manuscript.

---

## Author Comment (AC1)

Egusphere-2025-1950.pdf

Constraining microphysics assumptions on the modeling of Atmospheric Rivers using GNSS Polarimetric Radio Occultations

Authors: Antía Paz, Ramon Padullés, and Estel Cardellach

We sincerely thank the reviewer for their constructive and insightful comments, which have helped us improve both the clarity and robustness of our work. In this revised version, we have made several important changes to enhance the physical consistency and transparency of the methodology.

(1) We calrify that there is no direct coupling between ARTS and WRF for the main part of the analysis. Instead, ARTS is used diagnostically to evaluate which paritcle habits are most compatible with the WRF-derived water content and the best x-parameter obtained when comparing with actual observations of differential phase shift.

(2) Second, the optimization process has been refined and is now carried out at two levels: (i) within each microphysics scheme, to obtain the optimal set of x-parameters; and (ii) across all schemes, to identify the combination of microphysics and x-parameters that minimizes the cost function.

(3) In the comparison between the best x-parameter and those derived from ARTS, we now generate two distinct look-up tables that relate Kdp and WC, each corresponding to a different assumed particle size distribution (PSD). This modification ensures a closer alignment between the scattering properties used in ARTS and the microphysical assumptions in WRF.

**General**

The manuscript is novel and reports on unique way to assess the representation of hydrometeor habits in numerical weather prediction models.
Abstract and introduction. These sections start off with PRO technical details rather than "why" you are doing this work in the first place. Give a reason, some rationale. Suggest something like this:
Improving the prediction of precipitation intensity remains an elusive goal for the operational weather community. In current cloud resolving forecast models, the introduction of hydrometeors and the associated latent heat release are represented with either convective parameterizations or microphysical parameterizations or both, depending on the resolution of the simulation (Hristova Veleva et al 2021, https://doi.org/10.3390/atmos12020154). This study aims to assess the use of PRO in constraining the choice of microphysical assumptions within models, by exploiting the sensitivity of the PRO technique to the model's forecasted water content and selection of hydrometeor shapes.

The why of the analysis is now included both in the abstract and the introduction.

Section 2.3 is confusing as written with the "x-parameter" terminology. It's a strange sort of ad-hoc choice of terminology for a quantity that is fundamental to the material in the remainder of the article. If I understand your discussion, this is the "A" quantity in Eqn (6).

The A parameter in equation (6) is not the same as the x-parameters. Equation (6) accounts for the contribution of the liquid hydrometeor (rain), that is different from the one used for the frozen ones.

The rain contribution does not account for ARTS information. A and B are two constants that are the same for all the Atmospheric River cases.

**Specific**

**Near Line 30**. Suggest this wording: GNSS systems such as GPS transmit in a Right Hand Circularly Polarized (RHCP) state. A PRO-capable RO receiver uses dual orthogonal receive polarizations, horizontal (H) and vertical (V), enabling the measurement of differential phase delay ($\Delta\phi$, defined in Section 2.1 below). This differential delay is induced when the transmitted GNSS signals propagate through nonspherical hydrometeors (such as raindrops) in the atmosphere (Cardellach et al., 2015).

The paragraph was rephrased

**Line 54**. The work by Murphy et al 2019 was the original work that proposed investigating cloud resolving model microphysics. Also your work by Shu-Ya Chen fpt TC cases predates your study https://egusphere.copernicus.org/preprints/2025/egusphere-2024-3708/). Suggest this wording: Murphy et al (2019) proposed using PRO simulations to examine the model sensitivity to assumptions in its microphysical parameterization. However, PRO data were not available at that time. A recent study by Chen et al (2025, in review) compared observed and simulated ROHP data to examine a small number of tropical cyclones (TC) events. Our work focuses on the much larger number of PRO data from ROHP and Spire that cover Atmospheric River (AR) weather events. Together with PRO simulations, our comparisons aid in determining the extent to which variations in microphysical schemes in AR's can be discerned with PRO observations.

The new reference was added at the end of the section.

**Line 58**: However, the microphysical schemes implemented in mesoscale models such as WRF do not explicitly provide the required scattering characteristics at the GNSS frequencies (near 1.4 GHz) and limb-viewing angles needed to perform the Kdp computation.

The end of the paragraph was rewritten with the suggested wording.

**Line 66**. You need to cite the Hotta et al (2024) paper here, it was the first paper to simulate and study ROHP data in several AR cases. As they concluded, the AR cases provided the best agreement with ROHP owing to the nature of the weather event (more widespread, hence less variability along the long ray paths, relative to the TC cases with more variability from convective conditions). This is another reason why AR are good choices of weather events to examine.

The article was referenced.

**Near Line 95**. Most non-RO people will be unfamiliar with the term "ray path". The word is a construct used to refer to the volume of air that is sampled (discretely in time) by the PRO receiver as the RO rises or sets. I think a cross section figure would be helpful to explain what the path length is. You can provide one or refer to several in the Padulles et al papers, or cite the Figure from the summary paper from the "2023 Polarimetric Radio Occultations Workshop" in BAMS. https://doi.org/10.1175/BAMS-D-24-0050.1

In Figure 1 we have added a panel illustrating the occultation rays of that same observation.

**Figure 1**. The rays are from which PRO data? Perhaps make this for the case shown in Figure 7 so the reader has an idea of which rays sliced through the storm and where.

The figure was changed to one of the cases in figure 7.

**Figure 4 caption**. Be more specific, i.e., "Longwave IR brightness temperature image from geostationary satellite data".

The caption was changed.

**Line 180**. The term "x-parameters" is confusing. So, the x-parameters are the A and B terms in Eqn (6), for each level of the PRO profile, for each specie? (those relate Kdp and WC). Or are you fixing B for each microphysics type and letting only "A" be the "x-parameter". Explain this better as you refer to "x" a lot from this point on and is a source of confusion.

A and B do not depend on ARTS. They are constants for all the AR cases. The contribution for rain only depends on the WC, i.e. on WRF.

**Line 214**. Equation 8. One is used to seeing this type of equation for a data assimilation scheme in a forecast model. In that case, one can envision the "initial estimate" (the model background state). What is the background state (Xb) here, and how is it obtained? While y and H(x) are easy to envision- H(x) is your simulated $\Delta\Phi$ and y is the observed $\Delta\Phi$.

We thank the reviewer for this observation. In our case, we do not employ an initial estimate or "background state" as in classical data assimilation schemes. Instead, we only use the second term of Equation (8), since our approach does not rely on a first guess but rather performs a scan of the entire physically reasonable range of x-parameter values. To avoid confusion, we have removed the first term from the equation in the revised manuscript.

**Line 225**. From my understanding, you are assuming a diagonal background covariance matrix with all off-diagonal terms set to zero. Is correct, and if so, is this realistic? No need to change, just explain and justify.

No, we are not assuming a background covariance matrix in this analysis, as it is said in the manuscript. But we do consider a diagonal observation covariance matrix. While this is a simplification, it allows the optimization to be tractable and avoids introducing unverified cross-covariances. Since the focus of this article is to use PRO as a diagnostic tool, rather than a full assimilation system, we have considered this assumptions reasonable.

**Figure 6**. It's a stretch to see much of a relation here. Why not plot the integrated $\Delta\Phi$ above, say, the -10 C level (where the bulk of the ice is) instead? If there is a relation, it may be better revealed.

The aim of showing this kind of figure is that no general relation is observed between differential phase shift and water content. Instead, the relation will depend on the parameterizations and assumptions made for the different hydrometeors, and the specific atmospheric conditions of each case.

**Line 282**. You say, "the minimization of J is performed jointly for both the microphysics scheme and x-parameter". Should not this have been mentioned back when you introduced Eqn (8)?

Around Eq. 8 we have rephrased the paragraph explaining the optimization, to clarify this.

**Line 292**. You say, "Even slight variations in the snow's position can lead to differences in the results. However, the large-scale characteristics of atmospheric rivers tend to mitigate these effects."

For the first point, do you think that this is important, even for AR events? Cite the Hotta et al. 2024 manuscript for this last point.

We have clarified that, while the displacement of snow layers can indeed influence the results, the features of Atmopsheric Rivers structures mitigates the overall impact of such shifts in the optimizations. We now cite Hotta el al. (2024) to support this statement and highlight the robustness of AR large-scale characteristics against small positional errors.

---

## Author Comment (AC2)

**Review comments for Constraining microphysics assumptions on the modeling of Atmospheric Rivers using GNSS Polarimetric Radio Occultations**

We sincerely thank the reviewer for their constructive and insightful comments, which have helped us improve both the clarity and robustness of our work. In this revised version, we have made several important changes to enhance the physical consistency and transparency of the methodology.

(1) We calrify that there is no direct coupling between ARTS and WRF for the main part of the analysis. Instead, ARTS is used diagnostically to evaluate which paritcle habits are most compatible with the WRF-derived water content and the best x-parameter obtained when comparing with actual observations of differential phase shift.

(2) Second, the optimization process has been refined and is now carried out at two levels: (i) within each microphysics scheme, to obtain the optimal set of x-parameters; and (ii) across all schemes, to identify the combination of microphysics and x-parameters that minimizes the cost function.

(3) In the comparison between the best x-parameter and those derived from ARTS, we now generate two distinct look-up tables that relate Kdp and WC, each corresponding to a different assumed particle size distribution (PSD). This modification ensures a closer alignment between the scattering properties used in ARTS and the microphysical assumptions in WRF.

This paper investigates the potential of Polarimetric Radio Occultation (PRO) to provide insight into hydrometeor vertical structures, using WRF simulations and the ARTS scattering database to simulate differential phase ($\Delta\Phi$) for comparison with satellite observations. The use of PRO in this context is still emerging, and this study contributes to ongoing efforts to assess its sensitivity to different microphysical assumptions. Nonetheless, continued exploration of PRO for hydrometeor evaluation is valuable, and further development of physically consistent frameworks is encouraged.

That said, a fundamental concern arises from the physical inconsistency between the WRF microphysics (MP) schemes and the scattering properties derived from the ARTS habit database, which fundamentally undermines the interpretation and validity of the results.

1. Mismatch Between WRF MP Assumptions and ARTS Scattering Properties

The WRF MP schemes (e.g., Thompson, Morrison) are bulk microphysics parameterizations. They assume a fixed set of physical properties for each hydrometeor category: shape, density, and particle size distribution (PSD). For example, snow in the Thompson scheme may be represented by soft aggregates with a specific mass-size and PSD relationship, while graupel has a different assumed density and terminal velocity. These assumptions are not explicitly

output but are embedded in the diagnostic formulas that compute water content and number concentration.

In contrast, the ARTS habit database provides scattering properties (x-parameters) computed for discrete particle habits, each associated with its own shape, refractive index, and PSD assumptions. These may include bullet rosettes, dendrites, plates, spheres, or irregular shapes, and often with a PSD that differs from the one assumed in WRF MPs.

By using x-parameters from ARTS in combination with WC fields from WRF, the method effectively combines microphysical representations that were never meant to work together. This is a physical inconsistency — mixing the absorption/extinction/scattering characteristics of one assumed hydrometeor population with the mass distributions of another. Even if the x-parameters are bounded within ARTS values, this does not correct the mismatch in underlying particle physics.

**2. Optimization Does Not Evaluate Microphysics Skill**

The second issue concerns the interpretation of the optimization procedure. The authors optimize the x-parameters (within ARTS bounds) to minimize the difference between observed and simulated differential phase shift (Kdp), holding the WRF-derived WC fixed. This means that any biases or errors in the WC fields are effectively absorbed by tuning the x-parameters. Therefore, the optimization outcome does not evaluate whether the MP scheme is accurately predicting WC or phase shift — it only shows which x-parameters (within ARTS-defined bounds) can reconcile the WC fields with observations. This decouples the validation from the actual physical outputs of the MP scheme. Two different MP schemes could produce very different WC fields, but the optimization could find different x-parameters for each that yield similarly good fits to observations — misleadingly suggesting both are good, or that one is better based on fitted x alone.

**3. Misuse of "Forward Operator" Terminology**

The paper also uses the term "forward operator," which typically refers (in data assimilation) to a physically consistent transformation of model state variables (like WC) into observation space (like Kdp), based on known physics. In a proper forward operator, the mapping is fixed, and the state variables are adjusted (via a cost function) to minimize differences from observations.

However, in the present methodology, the forward operator is not fixed — it is being modified by optimizing x-parameters. This blurs the boundary between model physics and observation operator and makes it unclear what is actually being evaluated. Optimized x-parameters are not state variables and cannot be used to adjust the model state or improve forecasts, thus limiting the method's value even in a diagnostic context.

**4. Suggested Alternative**

A more physically meaningful approach would be to:
(i) Use x-parameters consistent with each MP scheme's assumed particle properties, either by matching ARTS habits or computing scattering from scratch;
(ii) Use those fixed x-parameters to compute simulated Kdp, and compare directly to observations, without optimization;

(iii) Evaluate the WC fields directly by assessing how well they reproduce observed Kdp under physically consistent scattering assumptions.

Summary
In short, while the idea of using observations of phase shift to evaluate model microphysics is important and timely, the current implementation is flawed due to the physically inconsistent blending of ARTS scattering properties with MP-derived WC, and an optimization procedure that does not test the microphysics predictions directly. The conclusions drawn about MP scheme performance are therefore not supported by the methodology, and a revision of the experimental design is recommended.

We thank the reviewer for this important point. We fully agree that the WRF microphysics schemes make their own assumptions about PSDs, densities, and particle shapes, which do not directly match the ARTS particle habits. As the reviewer suggests, one way to reduce this inconsistency would indeed be to adapt the PSD assumptions used in ARTS so that they reflect those in WRF. This is a valuable point that we have applied in the modified manuscript.

However, in our study we chose to keep the water content fields directly from WRF as input and then test which particle habits from ARTS are most consistent with the resulting PRO signal. Our goal is not to reproduce the exact internal assumptions of WRF schemes, but rather to ask: given the WC that WRF predicts, which types of particles would explain the observed PRO signatures? If we were to be 100% consistent with WRF microphysics, we would be restricted to spherical particles. In that case, the predicted $\Delta\Phi$ would be essentially zero because spherical particles do not produce differential phase shift. Such a test would not be meaningful for PRO, which is specifically sensitive to nonspherical hydrometeors.

Therefore, while we acknowledge the physical inconsistency highlighted by the reviewer, we emphasize that the strength of our approach is precisely testing wether water content output from WRF, when combined with realistic nonspherical particle habits, can reproduce the observed $\Delta\Phi$. This allows us to evaluate which microphysics schemes produce WC distributions that are most compatible with the scattering properties observed by PRO. We have clarified this reasoning in the revised manuscript and also added a discussion about the physical consistency and observational sensitivity. Also we have added a new reference of an study where they also couple WRF + ARTS and they demonstrate that the performance of the coupling has a good agreement with observations, [1].

[1] Wang, D., Prigent, C., Aires, F., & Jimenez, C. (2016). A statistical retrieval of cloud parameters for the millimeter wave Ice Cloud Imager on board MetOp-SG. *IEEE Access*, *5*, 4057-4076.

About the comment on the forward operator, it is true that looking for the best x-parameter may not fit within the strict definition of a forward operator. In this case, what we are doing is to use a simple, general operator that allows us to map WC into Kdp, and look for the best parameters that could be used in a potential operational forward operator. We can see in Hotta et al. 2023 that using fixed values for all cases may not lead to the right results, but rather that specific "x-parameters" may be better for specific phenomena (e.g. what works for

AR may not work for Tropical Cyclones). In this study, we define a way to infer such parameters, and we apply it to the case of Atmospheric Rivers.

Besides, there are some other comments related as follows:

L43: Passive microwave radiometers have also been utilized to interpret precipitation vertical structures, (Turk...
Comma here should be removed

Comma has been removed

L47: These change depending...
These changes depend

Corrected

L49: simulations will be conducted...
were conducted

Corrected

L100: As the PRO rays traverse the derived from PRO.
This is a broken sentence

Phrase deleted

L101: provides valuable insightse from GPS to LEO
Typo "insightse"

Corrected

L101: This, in turn, provides valuable insightse from GPS to LEO, refractivity gradients cause bending, resulting in rays becoming tangent to the surface at their lowest point, termed the tangent point, ht.
This whole sentence is not grammarly correct. Please fix.

Rephrase done

L107: matching a regular grid between 0 and 20 km
What was the reason to pick top height at 20km? Is this limit of the WRF model top?

The reason to choose 20km as the superior limit is because we do not expect the presence of clouds at those heights. The files include values until heights of 60km however, the 0.1km resolution is only for the first 20km.

L127: For some of the simulations, instead of using the RRTMG schemes for shortwave and longwave radiation, the New Goddard Shortwave and Longwave Schemes were used due to certain errors that occurred in specific simulations
Could these mis-match lead to any difference among MP results used in the following comparison?

This is a good point to remark. We are aware that by using different radiation schemes in some of the simulations we are not evaluating the microphysics over the same conditions. However, for each observation, the four different simulations varying the microphysics used the same radiation scheme so the comparison between them is not affected because the four simulations are done over the same conditions. The conditions then vary between observations, but since we analyze 37 cases and only a few of them are simulated with a different radiation scheme, we do not think that this mis-match could affect the statistics in a significant way.

The following is introduced in the manuscript to justify this mis-match:
"In a limited number of cases, the New Goddard Shortwave and Longwave radiation schemes were used in place of RRTMG due to technical issues. Both schemes are widely tested in WRF and provide consistent radiative forcing; since for each of the four simulations done for each observation the same radiation scheme is employed, and also since the amount of observations with this mis-match is low compared to the total amount of cases, we do not consider that this could affect in significance the statistics."

L139 whether hail or graupel is the third class of ice however,
A comma is needed before "however"
Section 2.2.1: The numbering of this section seems weird since there is no other subsection in section 2.2.

Comma corrected
Section 2.2.1 is now section 2.3

L215: where the x_b comes from? Is it from ARTS?
If so, please refer to my general comment

Equation (8) was written following the generic form of a variational cost function, but in our implementation we do not use a single firs-guess values x_b. Instead, the optimization explores values of x within predefined boundaries, as it can be seen in Figure 11. To avoid confusion, we have removed the first term from the equation in the revised manuscript.

L266: figure captions (a) and (b) are opposite to with what are defined in Figure 6. Caption

Corrected

L282: "Even though the minimization of J is performed jointly for both the microphysics scheme and x-parameter"
According to L210, x-parameters are obtained by optimizing the cost function as in (8) by fixing WC. Therefore, I don't understand what you meant as jointly.

By this we mean that both the optimization process is done at two levels, within a given microphysics scheme and overall (microphysics with lowest cost function evaluated at the optimal x values).

L293: The results indicate that the Goddard scheme consistently outperforms the others, followed by WSM6.
Since the simulation has used optimized x-parameters, which is case dependent (I think if there are other 32 points, you would get a different set of x-parameters), it is hard to say which original WRF simulation, and its corresponding scheme are the best. We can only say the Godard scheme was compensated by the specific "optimized" x_parameter in this particular case and such a combination performs the best.

We agree that, because of the x-parameters are optimized on a case by case basis, the evaluation cannot be interpreted as an absolute ranking of the original WRF microphysics schemes. Instead, the analysis should be understood as assessing how consistently each scheme, when combined with the scattering properties from ARTS, can reproduce the observed differential phase shift. In this sense, our conclusion that the Goddard scheme tends to align better with the observations reflects its relative performance within the optimization framework, rather than an absolute superiority of its microphysical assumptions.

Line 333. The first relevant thing to note is that the optimized values for ice and graupel often lay over the lower limit (the allowed range of x-parameters values is shown with a gray shaded area and related discussions).
This suggests the optimization problem is under-determined and constrained in a way that prevents a proper minimization of the cost function. As a result, I would question how robust the resulting conclusions are regarding the relative contributions of different hydrometeor species— especially if the optimization is not genuinely being achieved for some of them. The physical reason for not being able to minimize some of the MP results is that these MP schemes use different assumptions of the hydrometeor habits and therefore, you might not be able to get a convergence at all by using ARTS defined habits.
In its current form, the method raises a fundamental question about physical inconsistency between ARTS habits and MP assumptions. It might be helpful to examine whether the achieved x-parameters through Eq (8) match the assumption of the specific MP. If so, it might be meaningful to give some insights on habit distribution, matching the bulk parametrization but not described explicitly in MP. But it seems not the scope of current manuscript.

We indeed have made the optimization without boundaries, but the retrieved x-parameters for ice and graupel frequently became negative or reached unphysical values. This confirms that optimization problem is under-determined in some cases, and that unconstrained solutions do not provide physically meaningful results.

For this reason we choose to constrain the optimization by setting boundaries of the x-parameter to the minimum and maximum values found in the ARTS particle database. This ensures that the retrieved parameters remain within the physically plausible range of scattering properties. We are aware that these constraints may in some cases limit the cost

function minimization but we consider them necessary to guarantee physically realistic outcomes.

We again agree that this limitation also can reflect the mismatch between WRF and ARTS, but as we have said this should be interpreted not as providing a unique particle characterization, but rather as an indication of the relative tendency of each microphysics scheme to align with certain scattering behaviors.

---

## Author Comment (AC3)

**Summary**

This paper studies the Polarimetric Radio Occultation (PRO) technique to asses its sensitivity to vertical profiles of hydrometeors under varying microphysical assumptions in the context of atmospheric river cases. This sensitivity is theoretically explored using WRF model output from which differential phase shift $\Delta\Phi$ is simulated. The simulated $\Delta\Phi$ is compared against the observed with the aim to evaluate the applied microphysical schemes.

The study is well structured, clearly written and has informative figures. While I do think the study has the potential to be well-received, I have one major concern about the conclusions drawn from the optimization with the x-parameter method that is presented, in addition to some minor comments for clarity and quality improvements.

We sincerely thank the reviewer for their constructive and insightful comments, which have helped us improve both the clarity and robustness of our work. In this revised version, we have made several important changes to enhance the physical consistency and transparency of the methodology.

(1) We calrify that there is no direct coupling between ARTS and WRF for the main part of the analysis. Instead, ARTS is used diagnostically to evaluate which paritcle habits are most compatible with the WRF-derived water content and the best x-parameter obtained when comparing with actual observations of differential phase shift.

(2) Second, the optimization process has been refined and is now carried out at two levels: (i) within each microphysics scheme, to obtain the optimal set of x-parameters; and (ii) across all schemes, to identify the combination of microphysics and x-parameters that minimizes the cost function.

(3) In the comparison between the best x-parameter and those derived from ARTS, we now generate two distinct look-up tables that relate Kdp and WC, each corresponding to a different assumed particle size distribution (PSD). This modification ensures a closer alignment between the scattering properties used in ARTS and the microphysical assumptions in WRF.

**Main comments**

1. The method is based on x-parameters that relate water content (WC) to specific differential phase (KDP). Given the simulated WC, the 'optimal' x-parameter is then found by comparing simulated KDP against the observations. The authors conclude from this x-parameter that a specific particle habit is dominating the signal based on particle habits from the ARTS database. My concerns with this approach are 1): There will usually be a mixture of particles present, especially since measurements are done over a profile, and typically different particle habits dominate at different altitudes (temperatures). One example: Wouldn't a 50-50 mixture of particles with x-parameters of 0.1 and 0.3 yield an 'optimized' x-parameter of 0.2? In your current draft, you would then conclude that particles that relate to x-parameters of 0.2 are dominating. 2) The 'optimization' might lead to the correct results for the wrong reasons. E.g., a simulated water content that is much lower than in reality could be compensated by a higher x-parameter to achieve the correct KDP. 3) Keep in mind, that you 'overwrite' some of the particle properties that are used by the WRF microphysics schemes, by taking ARTS particle habits instead. For example, there are specific mass-size relations used, specific PSD shapes, and density assumptions. While I think this last point is

not a major problem, you should at least discuss it, since you goal is to 'evaluate' microphysics schemes.

(1) We agree that in reality the hydrometeor population within a profile is a mixture of particle types, and that different morphologies can dominate at different levels depending on temperature and growth regime. In the analysis, we are optimizing a cost function to get optimal x-parameters. Then, we obtain a range of values that generally represent best the observations, and are associated with specific particle types, but this is not a one to one link. The interpretation of the "dominant" particle should be understood as the effective habit or combination of habits that best explains the observations in a bulk sense, rather than as proof for a unique unique morphology.

Furthermore, because Atmospheric Rivers are large, spatially homogeneous events, and our study comprises 37 AR, much of the local variability of the particles is minimized. This helps ensure that the conclusions reflect robust large-scale behavior rather than being dominated by localized variability.

Finally, we view this work as a first step. For future work, we plan to refine the methodology by introducing a temperature-dependent operator, following the approach on [1]. We believe that event in its current form, the analysis is relevant enough to provide meaningful insights into the evaluation of cloud microphysics schemes.

[1] Kim, J., Shin, D. B., & Kim, D. (2024). Effects of inhomogeneous ice particle habit distribution on passive microwave radiative transfer simulations. *IEEE Transactions on Geoscience and Remote Sensing, 62,* 1-20.

(2) We are aware that a bias in the simulated WC could in principle cause an offset to the optimized x that yields Kdp leading to the same results, but for the wrong reason. Our analysis is framed in terms of relative comparisons among microphysics, all of which are run with the same WRF dynamical core and physical forcings. By doing the same for a significant number of events, we try to minimize the effect of biases in the WC fields.

(3) We are aware of the importance of this points. For this reason we have changed the methodology and calculated the x-parameters from ARTS employing the PSDs used in the different microphysics in order to be more coherent with the assumptions from WRF. In the way that this is done (i.e. finding the best x-parameter, and then assigning each parameter to a potential particle habit) the miss-match of the assumptions becomes less relevant. Also, being fully consistent with WRF assumptions is not possible, since the assumption of particles being shperes would invalidate the rest of the study (that is, perfectly spherical particles lead to 0 differential phase shift).

**Minor comments**

1. Line 100-104: I found it hard to understand this paragraph. Partly, because some of the sentences are incorrect. I also think a small sketch visualizing the ray-path and the position of $h_t$ on that path would help.

   This paragraph was rephased in order to make it more clear.

2. Line 105: I understand from this that the ray path is resolved with a given resolution. What is this resolution?

   Theryas are resolved with a resolution of 5km along the ray direction.

3. Line 115: Two-way or one-way nesting?

   We have employed two-way nesting in order to have more consistency across scales, to avoid mis-matches at the nest boundaries or to be more realistic regarding the large-scale evolution of the phenomena.

4. Line 120: What is the horizontal extend of the domain?

   The WRF simulations were configured with two nested domains. The outer domain uses a horizontal resolution of 15km with 130x146 grid points, giving an extent of approximately 1950x2190km. The inner domain has an horizontal resolution 3km with 466x526 grid points, corresponding to an extent of about 1398x1578km.

5. Line 127-128: Did you change the radiation scheme then for specific microphysics schemes only? Or for all microphysics schemes of that AR event?

   I have changed it for all the four schemes for one observation in order to be consistent between them when doing the comparison.

6. Line 134: Water vapor is typically not considered a 'hydrometeor'

   Corrected

7. Line 135: I would add here that the differences in assumed properties, such as particle density, are also important.

   Added
8. Line 153: Morrison is two-moment only for graupel, rain, snow and ice (not for cloud water). Also, Thompson predicts number concentrations (and thus, is two-moment) for cloud ice and rain. Here it sounds as if Thompson was completely one-moment.

   The paragraph was rephrased in order to be more clear.

9. Line 158: What were your conditions to define an AR event?

   We did not define the AR cases, we instead took them from a database [2] to see the coincident phenomena with our observations.

   [2] Guan, B., & Waliser, D. E. (2024). A regionally refined quarter-degree global atmospheric rivers database based on ERA5. *Scientific Data, 11*(1), 440.

10. Section 2.2: Was there any nudging applied?

No nudging (neither spectral nor grid nudging) was applied in our simulations. . All cases were initialized with ERA5 reanalysis and the run freely with the selected microphysics schemes.

11. Line 243: I don't fully understand the reasoning here. Isn't water content directly output by WRF? Why do you argue based on ΔΦ that snow is contributing the most?

Yes, the water content for each hydrometeor category is directly available from the WRF output. However, the key point here is that not all hydrometeors contribute equally to the differential phase shift. The contribution of each category depends not only on its water content but also on its scattering properties (x-parameter) and the geometry of the observation. In general the percentage of snow water content is greater in comparison to the rest of the hydrometeors, and in combination with the x-parameters the contribution is often the biggest.

12. Line 246: Could you see a height dependence? Or is that true for the full profile?

In some of the observations you can appreciate certain height dependence, however, we did not think that can be consider as a trend among the cases. For some of them in the first kilometers, the rain water content can achieve the same contributions as snow, whereas ice usually has the greater contribution above +-8km.

13. Line 256: How are you sure that Thompson is overestimating, and the other schemes are not underestimating snow water content?

Our interpretation that the Thompson scheme is "overestimating" snow water content is based on the fact that the resulting differential phase shift values in Thompson are consistenly higher than both the observations and the simulations from the other schemes, even when accounting for differences in particle scattering assumptions. However, we agree with the reviewer that this could also be interpreted as an underestimation of snow water content by the other schemes. We have added a new reference of an study where larger values of snow mixing ratio where found for the Thompson scheme in comparison with other schemes, like the WSM6.

14. Line 263-264: This sentence is confusing to me. Is there perhaps a word missing?

The sentence was rephrased for a better understanding.
15. Line 315: In my understanding that just means that the 'average' x-parameter that fits best is that of Rosette/Aggregate. See major comment 1).

See response to major comment (1).

16. Line 330: Is snow the largest contributor due to the largest x-parameter or largest WC, as indicated by Fig 7?

Snow appears as the largest contributor to the differential phase shift due to a combination of both factors: snow generally has comparatively large water contents in the simulated observations, and the x-parameters associated with snow habits are among the highest, reflecting their scattering efficiency.

17. Line 335: How is the error determined?

The error of the x-parameters is calculated in equation (11).

18. Line 364: Greater contribution to what. $\Delta\Phi$?

Yes, to the differential phase shift.

**Technical corrections**

1. Line 149: Brackes around the year only.

   Corrected

2. Figure 6: Panel (b): Colorbar label and Figure caption do not match. I think the figure caption for panel (b) is wrong.

   Corrected

3. Line 272: I think there is a word missing. Perhaps: ... no universal relationship ... **exists**, particularly...

   Sentence was rephrased

---

## Author Comment (AC4)

A new polarimetric radio occultation metric is used to assess the fidelity of different microphysical schemes in multiple simulations of atmospheric rivers. The results seem robust showing that xsnow~0.1 (aggregates) provides the best results and it is a good demonstration of a novel measurement that could be used to test other schemes and build climatologies.

This work should be publishable subject to satisfying the following points.

We sincerely thank the reviewer for their constructive and insightful comments, which have helped us improve both the clarity and robustness of our work. In this revised version, we have made several important changes to enhance the physical consistency and transparency of the methodology.

(1) We calrify that there is no direct coupling between ARTS and WRF for the main part of the analysis. Instead, ARTS is used diagnostically to evaluate which paritcle habits are most compatible with the WRF-derived water content and the best x-parameter obtained when comparing with actual observations of differential phase shift.

(2) Second, the optimization process has been refined and is now carried out at two levels: (i) within each microphysics scheme, to obtain the optimal set of x-parameters; and (ii) across all schemes, to identify the combination of microphysics and x-parameters that minimizes the cost function.

(3) In the comparison between the best x-parameter and those derived from ARTS, we now generate two distinct look-up tables that relate Kdp and WC, each corresponding to a different assumed particle size distribution (PSD). This modification ensures a closer alignment between the scattering properties used in ARTS and the microphysical assumptions in WRF.

**Main point.**

This work is introducing and demonstrating a novel metric to be used to test cloud microphysics representations. To provide context and convince the reader of its value it would be good (necessary?) to also provide comparison to more traditional metrics. Comparisons should be made to readily available satellite derived precipitation and top of atmosphere broad band radiation, perhaps even vapor and liquid water path too. Ideally these comparisons will also demonstrate the Goddard scheme performing best and supporting the result of the PRO analysis.

This would just form an extra section and add some paragraphs the results/discussion and conclusions.

We agree that such comparisons are valuable for building a comprehensive analysis of the model performance. However, the main goal of this work is to introduce and demonstrate the potential of PRO as a novel diagnostic for cloud microphysics, rather than to perform a full evaluation of the schemes against several available observational instruments.

To provide some context, we already include in Figure 3 and Figure 4 comparisons between the integrated vapor transport (IVT) and the accumulated total precipitation for the different schemes against ERA5 and also against longwave IR brightness temperature. Extending the analysis to a full set of additional metrics would require a full dedicated study and it is out of the scope of this analysis.

**Other points.**

line 24. I did not see anywhere a discussion about the horizontal resolution of this approach. That needs to be delved into along with a discussion about the pros and cons of using these long path lengths (~200km?)

The horizontal resolution in the along direction of the radio occultations is large, being a limb sounding technique and specially in comparison with other spacial techniques like infrared or passive microwave imagery. However, the rays can be resolved with very fine resolution, and when interpolated into the model grid, the horizontal resolution becomes less relevant.

It is true that since the contribution to the differential phase shift at each ray can come from everywhere along the ray, values along the ray can be compensated and there is an intrinsic ambiguity.

line 70. Perhaps the spatial coherence of these phenomena is also good for this approach that has coarse horizontal resolution?

Exactly, the selection of Atmospheric Rivers was made principally because of their spatial extension that algins well with the larger horizontal resolution of radio occultations.

line 84. This section would benefit from outlining the geometry of the sampling, perhaps with a schematic? Section 2.1 and 2.3 should also be merged? I read 2.1 and wondered why the ice phase was being ignored.

We have added on Figure 1 a second peanel showing the geometry of the occultation rays for that specific case.

Sections 2.1 and 2.3 are separated to make more easily the understanding of the methodology, first by introducing the concept of radio occultations and then to explain the observational operator used.

We have rephrase the sentence at the beginning of section 2.1 to make clear that we are considering also frozen hydrometeors.

line 127-128. What were the problems - it did not seem to be mentioned again.

The problem with RRTMG arose in a small subset of simulations where the model integration terminated prematurely due, from our prespective, numerical instability. This instability we think it was related to the coupling between the RRTMG radiation and microphysics in those particular cases, and we were unable to resolve it. We can not confirm the exact cause but we suspect that could be because of the presence of strong moisture transport.

line 140. Can probably just omit 'new'?

Omitted

line 158. Was a single instantaneous out chosen? Or was it averaged in some way. How close was the sample to the output in time?

We used the WRF output closest in time to the PRO observation. The temporal mismatch was always less than +-30 minutes. We did not average over multiple outputs.

line 170. See the main point. This paper can be strengthened substantially with some additional comparisons to satellite data to support the findings from this paper.

See response to the main point

line 227. The use of 50 assumes that each point is truly independent. Are these 'points' next to each other as in the gray area in figure 1? If so then they seem to be spatial coherent over scales of order 100km or so and lower number than 50 might be more appropriate?

No, this not assumes that each point is truly independent. The measurements each 1 second have influence of the 50 points next to it. The measurement frequency os of 50Hz.

line 248 and throughout. These schemes all use different size or mass thresholds to determine what is ice and what is snow. That could lead to big apparent differences between them that are not necessarily incompatible. For figure 5 do the schemes look more similar if ice+snow if plotted?

[Figure]

Correct, it kind of looks more similar for the four schemes. Indeed, the distinction between "ice" and "snow" varies among the parameterizations, since each scheme applies different size or mass tresholds to separate the two categories.

line 262-265. I was confused here. Equation 7 define iWC and delta Phi as being the integral along the limb sounding path. Here it talks about vertical integral? I am assuming that the WRF output is integrated along similar curved paths through the domain? How is that done?

The WC employed in the analysis is obtained from the WRF simulations. In order to obtain the Kdp by multiplying WC by the x-parameters, we first interpolate the WC obtained from WRF into the PRO rays, and then integrate the WC along each ray.

figure 6. All 4 schemes are represented - does this mean that each one was best in different cases?

Maybe i am misinterpreting.

Correct. For the different observations the best combination of WRF+ARTS yields to an effective microphysics, in general the one obtained for the majority of the cases is the Goddard however, for several others the rest of the schemes perform better.

Why not show all results - maybe a different panel for each scheme if it gets too messy?

We are showing the best performing scheme for each of the cases, that's why we choose to represent all of them in the same figure. Our goal is to show that no aparent relation exists. If we plotted them all, some of them would correspond to really bad fits, that is, maybe one point in the plot would be the best for a particular microphysics scheme, but with a really bad fit to the data. Then, these type of points would not be representative.

line 272. DeltaPhi is defined as dependent on WC and x. Does a surface regression against WC and x result in a better result?

It is true that other approaches, such as a surface regression of $\Delta\Phi$ against both WC and x, could be explored. However, since the WC is a simulated vairable and $\Delta\Phi$ is the observable, our goal is to design the analysis so that WC from WRF is kept fixed, while the x-parameters are adjusted within physically plausible bounds. This separation allows us to asses which microphysics schemes provide water content fields that are more compatible with the PRO observations, without directly fitting WC to $\Delta\Phi$. A regression including both WC and x would risk absorbing biases in WC into the fit, making it more difficult to know whether discrepancies arise from one part or the other.

line 278. But how can they all have the same WC and delta Phi axes in figure 7? The different x values will mean that each species should have its own delta Phi for a give WC?

There are two different x axes in Figure 7, one for the represented integrated water content and the other for the differential phase shift (corresponding to the PRO observation). As i can be seen in equation 4, yes, the different hydrometeors have different contributions to the differential phase shift.

figure 7. Maybe some more discussion about how the data is used is needed. It looks like you are plotting the curved limb path iWC value at its lowest point(?)

In Figure 7 we plot the integrated water content along each of the PRO rays vs the differential phase shift obtained from the PRO observation. Remember that iWC means integrated along the full ray-path (not vertical integral).

line 282. Can you tabulate J and the error. Perhaps indicate where the lowest J is significantly lower than the others?

The Appendix already contains a plot showing the cost function values, showing where the maximum and minimum values are encountered. For this reason, we have not included an additional table, since it would essentially duplicate the information already provided in graphical form. Moreover, the errors in Figure 11 are directly correlated with the cost function, which allows the reader to identify where the lowest values of the cost function coincide with the smallest errors

line 318-319. It would be useful to locate some ground based polarimetric studies of ARs to see if they agree with this result.

The study [1] was cited in relation with the WSM6 and Thompson schemes for their performance on Atmospheric River simulations.

[1] Jankov, I., Bao, J. W., Neiman, P. J., Schultz, P. J., Yuan, H., & White, A. B. (2009). Evaluation and comparison of microphysical algorithms in ARW-WRF model simulations of atmospheric river events affecting the California coast. *Journal of Hydrometeorology, 10*(4), 847-870.

---

## Author Response (AR2)

**Editor decision: Publish subject to minor revisions (review by editor)**

**Public justification (visible to the public if the article is accepted and published)**: Please revise the manuscript based on the reviewers' comments, and also update the verb tense throughout the text — for example, change "simulations were conducted" to "simulations are conducted." I noticed this type of error on pages 1, 2, 5, 6, 7, 10, 11, 12, 14, 16, and 20.

Reviewer's comments were addressed and the verb tense has been changed.

Also new citations were added in line 41 and line 50.

Anonymous Referee #4

I am happy to see that the authors responded thoroughly to the previously raised objections. I have only two minor comments, but no major issues remain. As such, I believe the manuscript is suitable for publication in Atmospheric Measurement Techniques, after addressing the minor comments below.

We thank the reviewer for their positive evaluation of our revised manuscript and for recognizing the improvements made in response to the previous comments. We appreciate the careful reading and constructive feedback provided throughout the review process. Below, we address the remaining comments and describe the corresponding revisions implemented in the manuscript.

Figure 1: The y-axis label of the left panel is not visible.

Thanks for noticing it, now it is visible.

Line 313: A citation supporting the known tendency of the scheme to overestimate snowfall (e.g., Jankov et al., 2009, as referenced later in line 337) should be provided here. Additionally, I recommend moving this sentence to line 305, where it would offer useful context for the comparison between the Thompson scheme and the others.

The citation was moved also to line 305, and sentence in line 313 was deleted since it is already mentioned in line 305 and 337.